# Engineered mRNA-expressed antibodies prevent respiratory syncytial virus infection

Pooja Munnilal Tiwari [1], Daryll Vanover[1], Kevin E. Lindsay[1], Swapnil Subhash Bawage[1], Jonathan L. Kirschman[1], Sushma Bhosle[1], Aaron W. Lifland[2], Chiara Zurla[1] & Philip J. Santangelo[1]

The lung is a critical prophylaxis target for clinically important infectious agents, including human respiratory syncytial virus (RSV) and influenza. Here, we develop a modular, synthetic mRNA-based approach to express neutralizing antibodies directly in the lung via aerosol, to prevent RSV infections. First, we express palivizumab, which reduces RSV F copies by 90.8%. Second, we express engineered, membrane-anchored palivizumab, which prevents detectable infection in transfected cells, reducing in vitro titer and in vivo RSV F copies by 99.7% and 89.6%, respectively. Finally, we express an anchored or secreted high-affinity, anti-RSV F, camelid antibody (RSV aVHH and sVHH). We demonstrate that RSV aVHH, but not RSV sVHH, significantly inhibits RSV 7 days post transfection, and we show that RSV aVHH is present in the lung for at least 28 days. Overall, our data suggests that expressing membrane-anchored broadly neutralizing antibodies in the lungs could potentially be a promising pulmonary prophylaxis approach.

[1] Wallace H. Coulter Department of Biomedical Engineering, Georgia Institute of Technology and Emory University, Atlanta, GA 30332, USA. [2] Institute of Bioengineering and Bioscience, Georgia Institute of Technology, Atlanta, GA 30332, USA. These authors contributed equally: Pooja Munnilal Tiwari, Daryll Vanover. Correspondence and requests for materials should be addressed to P.J.S. (email: philip.santangelo@bme.gatech.edu)

Acute respiratory infections are responsible for the hospitalization and deaths of millions of individuals annually worldwide[1]. Current vaccine strategies are limited to inactivated, recombinant, and live-attenuated types, though nucleic acid vaccination is actively being investigated in preclinical and clinical trials[2–8]. These approaches, though very powerful, take time to take effect, limiting their utility during pandemics. Additionally, prophylaxis remains limited to antivirals and, in the case of respiratory syncytial virus (RSV), the broadly neutralizing antibody palivizumab is the only FDA approved treatment for high-risk populations. The limited application of palivizumab is likely due to its debated efficacy. Palivizumab, delivered by intramuscular (IM) injection, is present in the serum at titers 2000 fold higher than in bronchoaveolar-lavage (BAL) samples[9]. This indicates that the majority of palivizumab injected IM is not delivered to the appropriate organ compartment to neutralize the virus, potentially explaining the limited reduction in hospitalization rates observed in treated infants[10]. Therefore, there is a need for a rapidly-expressed, targeted prophylaxis technique to prevent pulmonary infections.

To date, no DNA-based therapeutics have been approved for human use due primarily to safety concerns. First, DNA-based therapeutics incite concerns of integration into the host genome. While this effect has been minimal to date, integration must continue being monitored for each antigen expressed by the DNA[11,12]. Modified adeno-associated viruses (AAV) are a prominent vehicle for nucleic acid therapeutic delivery, in part due to being replication deficient and minimally pathogenic[13–15]. However, because AAV-based therapies are permanent and elicit an immune response, repeated dosing is not currently possible[16–18]. Moreover, this immune response precludes the ability to deliver additional AAV-based therapeutics targeting different pathogens or additional strains.

Delivery of therapeutic-encoding mRNA directly to the lung aims to safely and transiently increase therapeutic proteins in the target organ, compared to systemic purified recombinant protein delivery, frequently administered IM[19–21]. Additionally, delivery of naked mRNA produces more protein during peak expression than naked plasmid DNA[22]. Still under investigation, aerosol delivery of mRNA has been shown to elicit transient protein expression capable of treating disease[21,23,24]. Targeted delivery of therapeutics to the organ of interest has the potential to minimize systemic toxicity, anti-antibody immune responses, and reduce the amount of drug required to achieve therapeutic levels.

Here, we developed a modular toolbox to express synthetic, modified mRNA and prevent viral infections in the lung. First, we expressed whole palivizumab (secreted, termed sPali) in the lung via synthetic mRNA delivery by intratracheal aerosol. Second, we linked the well-characterized glycosylphosphatidylinositol (GPI) membrane anchor sequence from the decay accelerating factor (DAF) to the palivizumab heavy chain mRNA (Fig. 1a, b)[25,26]. Anchored palivizumab was termed aPali; we hypothesized that cells transfected with aPali would retain the immunoglobulin on the epithelial surface, increasing its concentration in the lung and improving efficacy. Finally, we demonstrated that the GPI anchor is adaptable to other constructs by linking it to a RSV-neutralizing VHH camelid antibody (RSV aVHH), previously demonstrated to be more potent than palivuzmab[27].

In this study, we find that mRNA-expressed anchored neutralizing antibodies, both whole and single domain: (1) are rapidly expressed and retained on the surface of transfected cells in culture and in mouse lungs, (2) prevent RSV infection in vitro and in vivo, and (3) are minimally inflammatory. Along with aPali and sPali, we also demonstrate that anchored, but not secreted, single domain antibodies have improved in vivo half-life, preventing RSV infection 7 days post transfection. Together,

these technologies comprise a modular mRNA toolbox to express a variety of engineered neutralizing antibodies to prevent pulmonary infections.

## Results

**aPali is anchored to the membrane and inhibits RSV in cells.** We first verified the assembly of whole antibodies expressed from synthetic mRNA by staining with an anti-human IgG antibody in vitro. The anti-human IgG antibody efficiently stained cells co-transfected with mRNAs encoding the light and heavy chains at a 1:4 molar ratio. No staining was observed in cells transfected with only light chain mRNA, while transfection with heavy chain mRNA resulted in significantly lower staining. Therefore, we concluded that mRNAs for both heavy and light chains were efficiently translated to form whole IgG (aPali) (Fig. 1c).

Next, to establish that the GPI membrane-anchor encoded in the heavy chain sequence was functional, we transfected cells with mRNA encoding for either aPali or sPali, and compared the localization of assembled antibodies at 2, 4, 6, 12, and 24 h post transfection, with and without permeabilization (Fig. 1d). Without permeabilization, strong surface staining was visible as early as 2 h in cells transfected with aPali mRNA, indicating rapid antibody production and localization to the plasma membrane. The surface staining was strongest between 6 and 12 h after transfection and remained through 24 h after mRNA delivery. Because the aPali concentrations in the secretory membrane system were high at early time points, the contrast was reduced compared to the images in Fig. 1c (Supplementary Fig. 1). In contrast, in cells expressing sPali mRNA, membrane staining was minimal, indicating that expressed antibodies were not localized to or concentrated on the plasma membrane.

When the localization was evaluated with permeabilization, sPali was found in the secretory membrane system 2 to 12 h after transfection, with a dip in intensity at 24 h, possibly due to both the decreased mRNA expression and diffusion of sPali into the media. Interestingly, aPali was observed in the secretory membrane pathway 2 to 6 h after transfection, but at lower concentrations, indicating a reduced accumulation in this cell compartment, as well as on the plasma membrane as early as 2 h post-delivery. Together, these data show that the GPI membrane-anchor is functional and anchoring the antibody on the membrane prevents aPali diffusion into the growth media.

To demonstrate that aPali can bind and inhibit RSV infection, we transfected Vero cells with mRNA encoding either the heavy chain, light chain, or both the heavy and light chains. The cells were infected with RSV 24 h after mRNA delivery; infected cells not transfected with mRNA were used as a positive control. Cells were then stained for aPali, RSV nucleoprotein (RSV N), and with a panRSV polyclonal antibody (Fig. 1e). No infection was detected in cells transfected with both the heavy and light chain of aPali. This demonstrates that transfection with both chains is required for successful binding and neutralization of RSV. This was quantified by measuring the volume of RSV N signal upon infection (Fig. 1f). In untransfected cells, RSV replicated, with the RSV N signal representing approximately 21% of the total volume per cell. In aPali transfected cells, we detected a 99% reduction in RSV N volume, with the residual signal likely derived from the viral particles in the inoculum. To demonstrate that this effect was not specific to Vero cells, we transfected aPali and sPali mRNA into A549 cells, a human lung epithelial cell line, again showing that whole antibody is required for RSV inhibition (Supplementary Fig. 2). In addition, aPali mRNA transfection resulted in a 2-log reduction of RSV titer via plaque assay (Fig. 1g,

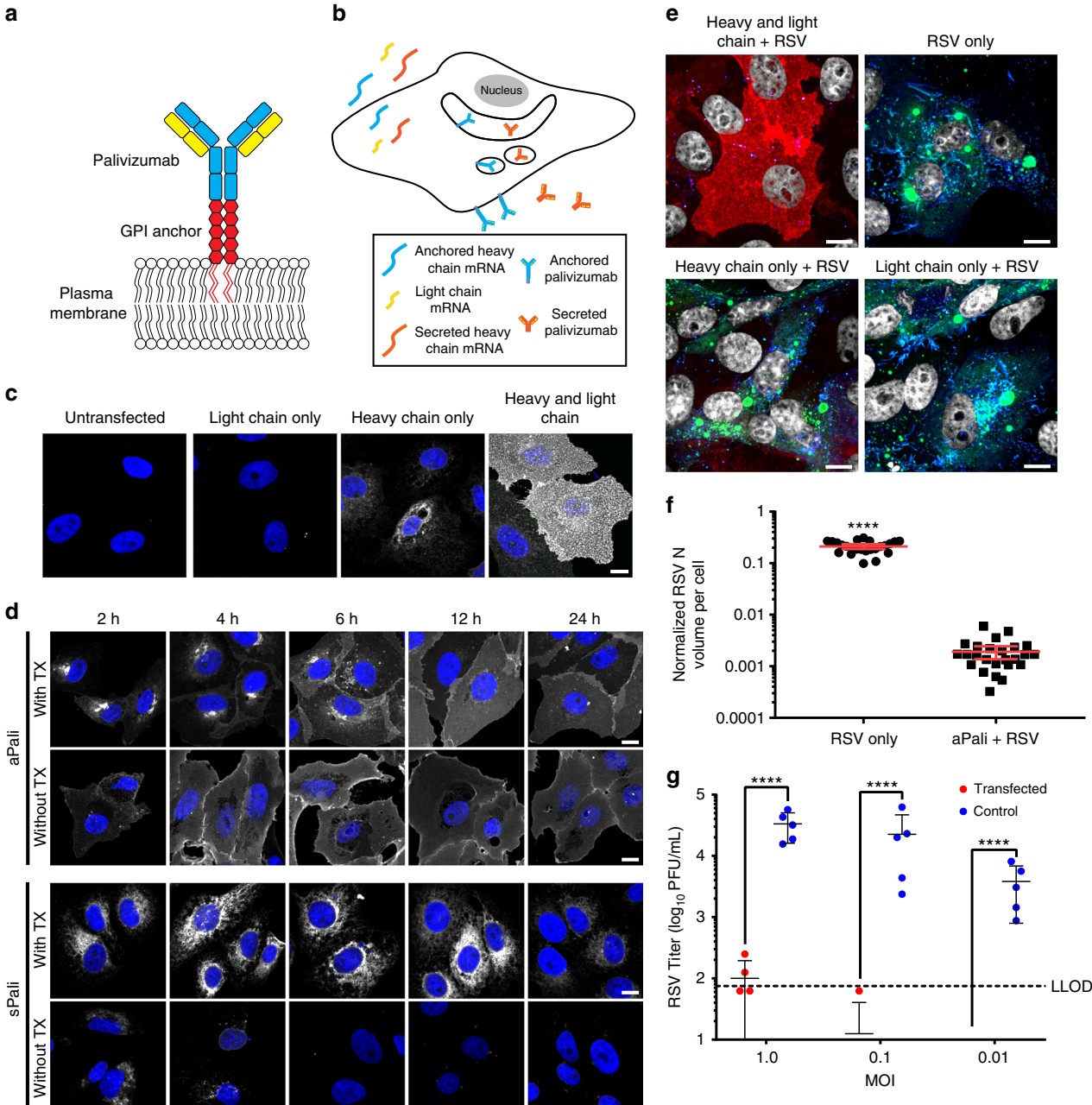

**Fig. 1** aPali is anchored to the membrane and inhibits RSV in cells. **a** Schematic of aPali anchored to the plasma membrane. **b** Schematic of aPali and sPali mRNA delivery and expression. **c** Cells were transfected with vehicle control or 1 μg of either aPali heavy chain only, aPali light chain, or both the heavy and light chain mRNAs. 24 h later, cells were fixed, permeabilized, and stained with a donkey anti-human secondary antibody (white). Scale bar represents 10 μm. **d** Cells were transfected with 1 μg of either aPali or sPali mRNA. At 2, 4, 6, 12, and 24 h, cells were fixed, permeabilized or not, and stained for the expressed antibody (white). Scale bar represents 10 μm. **e** Cells were transfected with vehicle control or 1 μg of either aPali heavy chain only, aPali light chain only, or both the heavy and light chain mRNAs. After overnight incubation, cells were infected for 24 h before being fixed and stained for RSV N (green), panRSV (blue), and aPali (red). Scale bar represents 10 μm. **f** Quantification of the mean volume of the RSV N signal per cell from microscopy images in part (**e**). Error bars represent 95% confidence intervals. Asterisks indicate $p < 0.0001$ (Mann–Whitney U test). 25 cells were analyzed per group. Results represent mean of two independent experiments. **g** Cells were transfected with vehicle control or 1 μg of aPali and infected with RSV at MOI of 0.01, 0.1, and 1. Supernatants were collected at 24 hpi and virion titers were measured by plaque assay. Each data point represents mean of 4 replicate wells. Error bars represent standard deviation. Dotted line represents lower limit of detection. Asterisks indicate $p < 0.0001$ (one-way ANOVA of log-transformed data). Results represent mean of three independent experiments

Supplementary Fig. 3). To verify that transfection of a lower dose of aPali mRNA does not enhance RSV infection, we performed a titration of aPali mRNA from 100 to 2000 ng (Supplementary Fig. 4). We found that all concentrations of aPali mRNA significantly inhibited RSV replication, compared to either mock or irrelevant mRNA control treated cells, in a dose-dependent manner. Overall, these data demonstrate that aPali mRNA transfection prevents RSV infection in vitro.

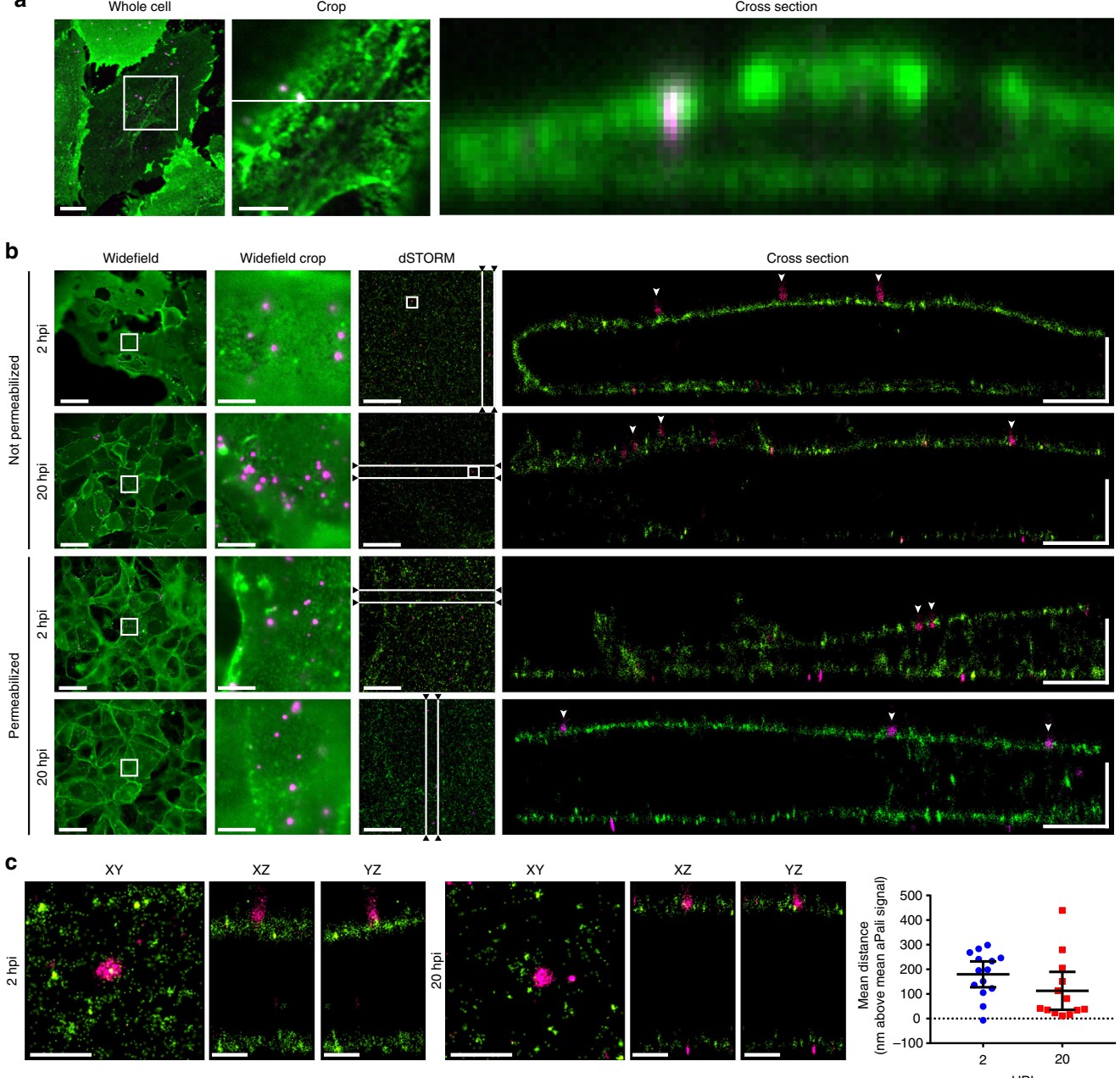

**Fig. 2** Mechanism of action of aPali by super-resolution microscopy. **a** Vero cells were transfected with 1 µg of aPali and infected with RSV at MOI of 1. At 24 hpi, cells were fixed, stained with anti-human secondary (green) and for RSV N (magenta) and imaged using a confocal microscope. Cropped image is a magnification of white box inset. Cross section is along the white line in the cropped image. Scale bar in whole cell represents 10 µm, while scale bar in cropped image represents 5 µm. **b** The same cells from part (**a**) were imaged using 3D dSTORM. Cells were fixed at either 2 or 20 hpi and permeabilized or not, as indicated. Widefield reference images were captured on the same microscope immediately prior to acquisition, with the cropped region indicating magnification of the white box. dSTORM images were acquired on the widefield cropped region. Cross section views are shown from a 2 µm slice through the XY image as indicated by the white lines. Arrowheads indicate single RSV virions. Scale bar in the widefield images represent 20 µm, while scale bars in the widefield crop and dSTORM images represent 5 µm. Scale bar "L" represents 2 µm in both directions. **c** Single RSV particles were cropped from white boxes in the dSTORM images from unpermeabilized cells in part (**b**). XY views, as well as XZ and YZ are shown. Scale bars represent 700 nm. A 1 µm square ROI was centered on individual viral particles and the mean distance in the *z* direction between the aPali and RSV N signals per particle were determined at 2 and 20 hpi ($n = 13$). Error bars represent 95% percent confidence interval. No significant difference was found, $p > 0.05$ (Mann–Whitney U Test). Results represent mean of two independent experiments

**Mechanism of action of aPali by super-resolution microscopy.** Next, we sought to delineate the mechanism by which aPali prevented infection of transfected cells. Conventional spinning disk confocal microscopy does not afford the necessary axial resolution to determine if RSV particles are internalized in aPali transfected cells or merely attached to the plasma membrane (Fig. 2a). To overcome the resolution constraint, we employed a 3D direct stochastic optical reconstruction microscopy (dSTORM) system with ~20 nm lateral and ~50 nm axial resolution. We transfected cells with aPali mRNA

overnight before infecting them with RSV. At 2 and 20 hpi, cells were immunostained for aPali and RSV N, with and without permeabilization. In RSV only cells, large inclusion bodies and long filaments were visible (Supplementary Fig. 5). In transfected cells, using dSTORM, we observed aPali at the plasma membrane along the contour of the cell (Fig. 2b). Additionally, at 2 and 20 hpi, individual RSV virions were observed at the aPali-labeled membrane. Specifically, at both time points, single virions of ~100–300 nm, the expected size of spherical-like RSV particles, were localized 180 nm (at 2 hpi) and 113 nm (at 20 hpi) above the plasma membrane (Fig. 2c). These results revealed that RSV particles were not internalized over time, demonstrating that the mechanism by which aPali inhibits infection is by preventing fusion and cytosolic uptake of RSV.

**Distribution of aPali mRNAs and expression in lung epithelia.**
Before performing in vivo challenge studies, we determined the distribution of aPali mRNA in the lung. To accomplish this, we used fluorescent multiply labeled RNA imaging probes (MTRIPs) bound to the mRNA prior to delivery[28,29]. Recently, we demonstrated that this method allowed for the tracking of exogenous mRNA using a variety of detection techniques while not significantly impacting translation[30]. We labeled aPali heavy chain mRNA with DyLight-650 MTRIPs and aPali light chain mRNA with Cy3B MTRIPs and delivered the labeled mRNA into the lungs of BALB/c mice by intratracheal aerosol. After 4 h, heavy and light chain mRNA distributions in the lung were analyzed by confocal microscopy. In transfected lungs, mRNA was distributed throughout the tissue (Supplementary Fig. 6a). High-magnification images confirmed that both heavy and light chain mRNAs colocalized in discrete granules inside transfected cells (Fig. 3a). We then analyzed the colocalization of the exogenous mRNA with either Rab5, an early endosome marker, or Rab7, a late endosome marker. We found that 24.4% of delivered mRNA colocalized with late endosomes and 39.7% with early endosomes, indicating that some of the mRNA was still trafficking through the endosomal system of alveolar cells 4 h post transfection (Fig. 3b, Supplementary Fig. 6b). Since 75.6% and 60.3% of mRNA was not colocalized with Rab 5 or Rab 7, respectively, a significant fraction of this mRNA was likely free within the cytosol (Fig. 3c)[30].

Next, we assessed whether aPali mRNA was expressed and retained on the surface of transfected cells in vivo. We delivered aPali or sPali encoding mRNA using Viromer RED (VR), a modified form of PEI, to the lungs of BALB/c mice. After 4 h, lung tissue sections were stained for the expressed antibody and imaged (Fig. 3d). Lungs transfected with aPali displayed strong antibody signal on the apical surface of the airway epithelial cells. On the contrary, lungs transfected with sPali showed pockets of expressed antibody predominately within alveolar epithelial cells. Then, we evaluated the distribution of DyLight-680 labeled palivizumab injected via IM and studied the distribution of the antibody in vivo using a handheld near-infrared imager and microscopy. We found that DyLight 680-labeled antibody was present in the lungs and throughout the body, with the most intense fluorescent signal in the injected tibialis muscle (Supplementary Fig. 7). Next, we sought to quantify the percentage of lung cells expressing aPali upon intratracheal aerosol. We transfected mice with mRNA encoding the aPali heavy chain and a light chain sequence modified with a C-terminus V5 peptide tag, for detection via flow cytometry. After 24 h, mice were sacrificed, and the lungs were analyzed for V5 expression. We found that in two of the transfected mice, approximately 45% of the total isolated lung cells were expressing

aPali-V5 (Fig. 3e). One of the three mice had minimal V5 staining, likely due to variability in the intratracheal aerosol delivery. This indicates that aPali can be expressed in mice lungs and is anchored to the surface of transfected cells in the lungs.

**Optimizing mRNA delivery vehicle and dosage.** Due to limitations in the mouse model for RSV, we evaluated the effect of a protein kinase R (PKR) inhibitor, C16, on RSV infection. When mock transfected mice given saline and C16 were infected 1 day later, we obtained significantly higher RSV F copy number (Supplementary Fig. 8a). We then assessed RSV titers at day 1 and 4 post infection by qRT-PCR. At 4 dpi, the difference between aPali mRNA transfected and control infected animals was more pronounced (Supplementary Fig. 8b). These results are consistent with previous reports of L19 RSV infections in mice[31]. Finally, we confirmed that aPali mRNA inhibits RSV replication irrespective of C16 addition by transfecting mice with either saline or aPali mRNA, with or without C16, and infecting the mice 24 h later. At 4 dpi, we measured RSV F copies, finding that aPali significantly inhibits RSV replication in either case (Supplementary Fig. 8c). To assay the effect of C16 on PKR activation, we assessed eIF2α phosphorylation, a downstream effect of PKR activation, by ELISA and western blot. We detected no differences in eIF2α phosphorylation, indicating that PKR activation was not significantly potentiated by either mRNA delivery or presence of C16 (Supplementary Fig. 8d, e).

To optimize the delivery of mRNA to the lung, we compared several mRNA formulations and doses. We first aimed to find the optimal delivery vehicle for aerosol-mediated transfection into the lungs which would yield effective inhibition of RSV infection. We evaluated several commercially available vehicles including VR and in vivo-jetPEI, both PEI derivatives, as well as naked mRNA delivery in water. We transfected mice intratracheally with 40 μg of aPali mRNA, and, after 24 h, infected them with RSV L19 intranasally. All three delivery vehicles significantly reduced RSV F copy numbers compared to the mock-transfected (saline only), RSV infected control (Fig. 4a). Specifically, naked mRNA and mRNA delivered by VR reduced RSV F copy number by 91% and 96%, respectively. Additionally, only naked and VR delivered mRNA significantly reduced RSV viral titers in lung homogenate supernatants, by 75.4% and 74.2%, respectively (Fig. 4b). To determine the immunological effects of the delivery vehicles and mRNA, we measured the mRNA levels of IL-6, CCL5, and interferon gamma (IFNg) and beta (IFNb) and found that in vivo-jetPEI increased IFNb mRNA levels by 257-fold, commensurate with a significant decrease in body weight on days 1 and 2 post transfection (Fig. 4c, Supplementary Fig. 9). The increase in IFNb mRNA levels indicated an increase in immune activation, absent when using VR or naked mRNA delivery. Finally, to investigate the extent of RSV infection, we also immunostained lungs from each group using a pan-RSV polyclonal antibody. While both the RSV control and in vivo-jetPEI groups yielded multiple pockets of infection throughout the lung, minimal RSV staining was observed when aPali mRNA was delivered either by VR or naked, with the latter having the more dramatic impact (Fig. 4d). The reduced staining in the VR and naked images is clearly due to the cell-associated virus being detected via microscopy. Our plaque assay protocol predominately measures free virions, which could be from the inoculum. To minimize immunological risk, maximize prevention of RSV infection, and increase translational suitability, naked mRNA delivery was used for subsequent experiments.

Next, we optimized the mRNA dose by delivering 20 or 100 μg of aPali mRNA. To compare the therapeutic effect of aPali mRNA to injected palivizumab, we also administered 15 mg/kg (300 μg

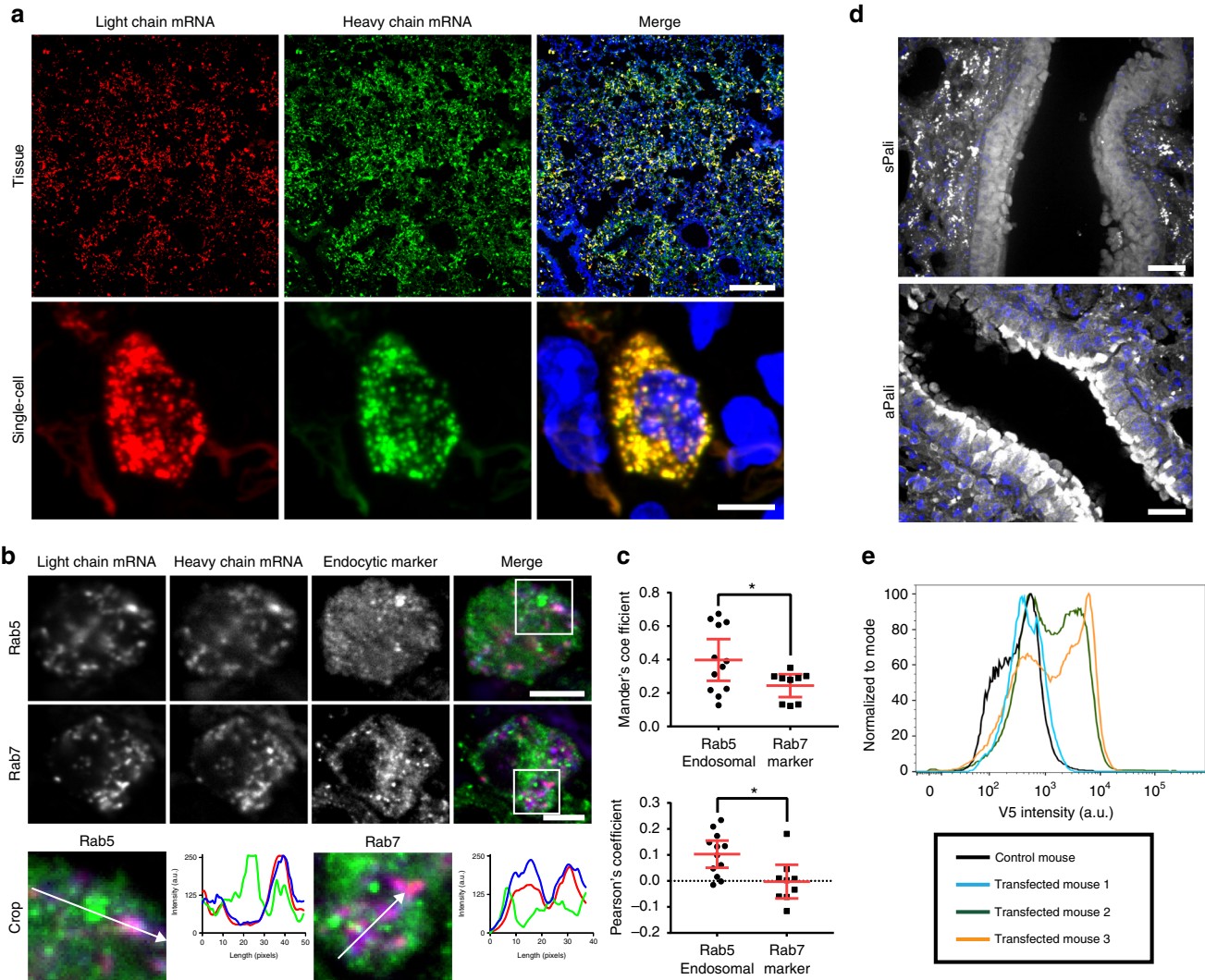

**Fig. 3** Distribution of aPali mRNAs and expression in lung epithelia. **a** Cy3B labeled light chain (red) and DyLight 650 labeled heavy chain (green) mRNA was transfected into the lungs of mice. At 4 h, lungs were sectioned and imaged on a whole tissue (top) or single cell basis (bottom). Scale bar represents either 200 μm (top) or 5 μm (bottom). **b** Tissue sections from part (**a**) were stained for Rab5 or Rab7 (green) for early or late endosomes, respectively. Scale bar represents 5 μm. Cropped regions are magnifications of white boxes in whole cell images with intensity profiles along the direction of the white arrow. **c** Mander's colocalization coefficient and Pearson's correlation between light chain mRNA and Rab5 or Rab7. Dotted line indicates 0 correlation. $n \geq$ 10 cells per group. Error bars represent 95% confidence interval. Asterisk indicates $p < 0.05$ by two-tailed $t$-test. **d** Mice lungs were transfected with either sPali (top) or aPali (bottom) mRNA using Viromer Red. At 4 h, tissue sections were stained for the expressed antibody (white). Scale bar represents 25 μm ($n = 2$ mice per group). **e** Mice lungs were transfected with aPali heavy chain and V5 tagged light chain mRNA. Flow cytometry was performed on dissociated lungs stained for V5. Histograms of V5 intensity were normalized to the mode of intensity

for a 20 g mouse in 25 μl of saline) of antibody by IM injection into the tibialis muscle. At 4 dpi, mice were sacrificed and lungs were excised, processed for mRNA extraction, and total RSV F mRNA copy number was measured by qRT-PCR. We found that 100 μg of aPali mRNA significantly reduced RSV F copies by 90% and 87% compared to the RSV control and irrelevant mRNA controls, respectively, while 20 μg of aPali mRNA yielded no significant decrease in RSV F copy number compared to the RSV control (Fig. 4e).

**mRNA-expressed palivizumab prevents RSV infection in vivo.** We then compared the RSV-neutralizing ability of aerosolized aPali mRNA and sPali mRNA (100 μg) to IM palivizumab (300 μg) and RSV controls, given prophylactically. mRNA encoding for either aPali HC (heavy chain) only or Flu aVHH was delivered

as an irrelevant mRNA control. 24 h post-delivery, mice were infected with RSV L19. Mice were sacrificed and lungs excised at 4 dpi and processed for cryosectioning, protein and virus collection, and RNA extraction. We found that aPali and sPali mRNA significantly reduced viral titers by 65% and 89%, respectively (Fig. 5a), when supernatants were analyzed for virion production. We also detected significantly fewer plaques in infected animals treated with either aPali or sPali mRNA compared to control animals treated with irrelevant mRNAs (aPali heavy chain only or Flu aVHH). As expected, palivizumab IM yielded no detectable plaques due to free antibody neutralization of RSV virions directly in the supernatant. All treatments significantly limited RSV replication in the lung (Fig. 5b), reducing RSV F copies by approximately 90% compared to RSV only animals. Animals treated with aPali or sPali mRNA and infected

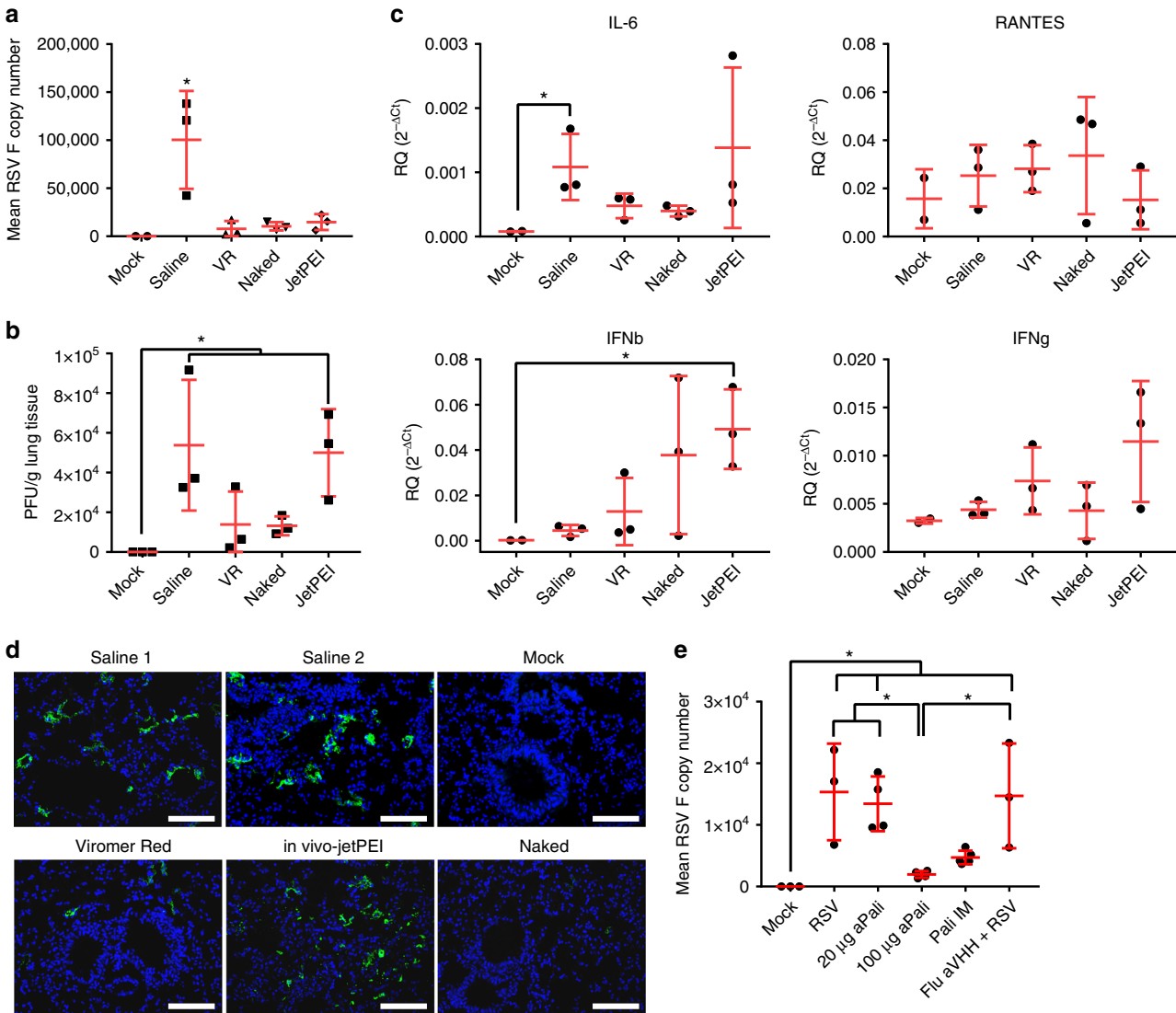

**Fig. 4** Optimizing mRNA delivery vehicle and dosage. **a** Mice lungs were transfected with 40 μg of aPali mRNA using the indicated delivery vehicles. Saline and negative controls are indicated. 24 h later, mice were infected with RSV L19. At 4 days post infection, mice were sacrificed and lungs were excised and processed for mRNA extraction. Mean RSV F copy number was determined by qRT-PCR. Error bars represent standard deviation. Asterisk indicates p < 0.05 (one-way ANOVA, Tukey's multiple comparisons). **b** The same mouse lungs from part (**a**) were processed and RSV virions were measured by plaque assay. Error bars represent standard deviation. Asterisk indicates $p < 0.05$ (one-way ANOVA, Tukey's multiple comparisons). **c** Mice lungs from part (**a**) were analyzed by qRT-PCR for the relative quantity of mRNA for IL-6, CCL5, IFNg, and IFNb. Error bars represent standard deviation. Asterisk represents p < 0.05 (Kruskal–Wallis, Dunn's multiple comparisons). **d** Mice lungs were transfected as in part (**a**). Lungs from two different saline group animals are included for clarity. Lungs were processed for cryosectioning and stained with panRSV antibody (green). Scale bar represents 100 μm. **e** Mice lungs were transfected with 0, 20, or 100 μg of aPali mRNA or 100 μg of Flu aVHH mRNA, as a control. Palivizumab injection IM was included as a positive control. 24 h later, mice were infected with RSV L19. At 4 dpi, mice were sacrificed and lungs were excised and processed for mRNA extraction. Flu aVHH mRNA group data was derived from a separate experiment. Mean RSV F copy number was determined by qRT-PCR. Error bars represent standard deviation. Asterisk indicates $p < 0.05$ (one-way ANOVA, Tukey's multiple comparisons)

had significantly reduced RSV F copies compared to control animals treated with two different irrelevant mRNAs. Given, the irrelevant control mRNAs had no effect on F copy numbers, additional animals were not sacrificed for microscopic evaluation, as the data did not justify their use.

Next, the RSV infections and treatments were evaluated microscopically. We found that while bright pockets of RSV infection throughout the tissue were visible in the RSV control group, aPali and sPali mRNA delivery significantly reduced the RSV staining (Fig. 5c, Supplementary Fig. 10); specifically, aPali reduced the volume of RSV signal by more than 70% (Fig. 5d, e). The safety of mRNA delivery to the lung was also assessed by

measuring cytokine protein concentrations 5 days post transfection using a multiplexed immunoassay. None of the measured cytokines, CCL5, IL-12 p70, CCL3, and IL-6, were statistically different from mock controls (Supplementary Fig. 11). These results are consistent with the report that maximal cytokine response to RSV L19 infection occurs at 8 dpi, while we sacrificed mice at 4 dpi to obtain maximal viral titer[31]. To evaluate cytokine expression in response to mRNA delivery, we transfected mice with aPali, sPali, or RSV aVHH mRNA diluted in water only (without C16). When we assayed the lungs of these mice at 24 h post transfection, without infection, we detected no significant difference in the levels of: IL-12 p70, IL-6, TNFα, or IFNα

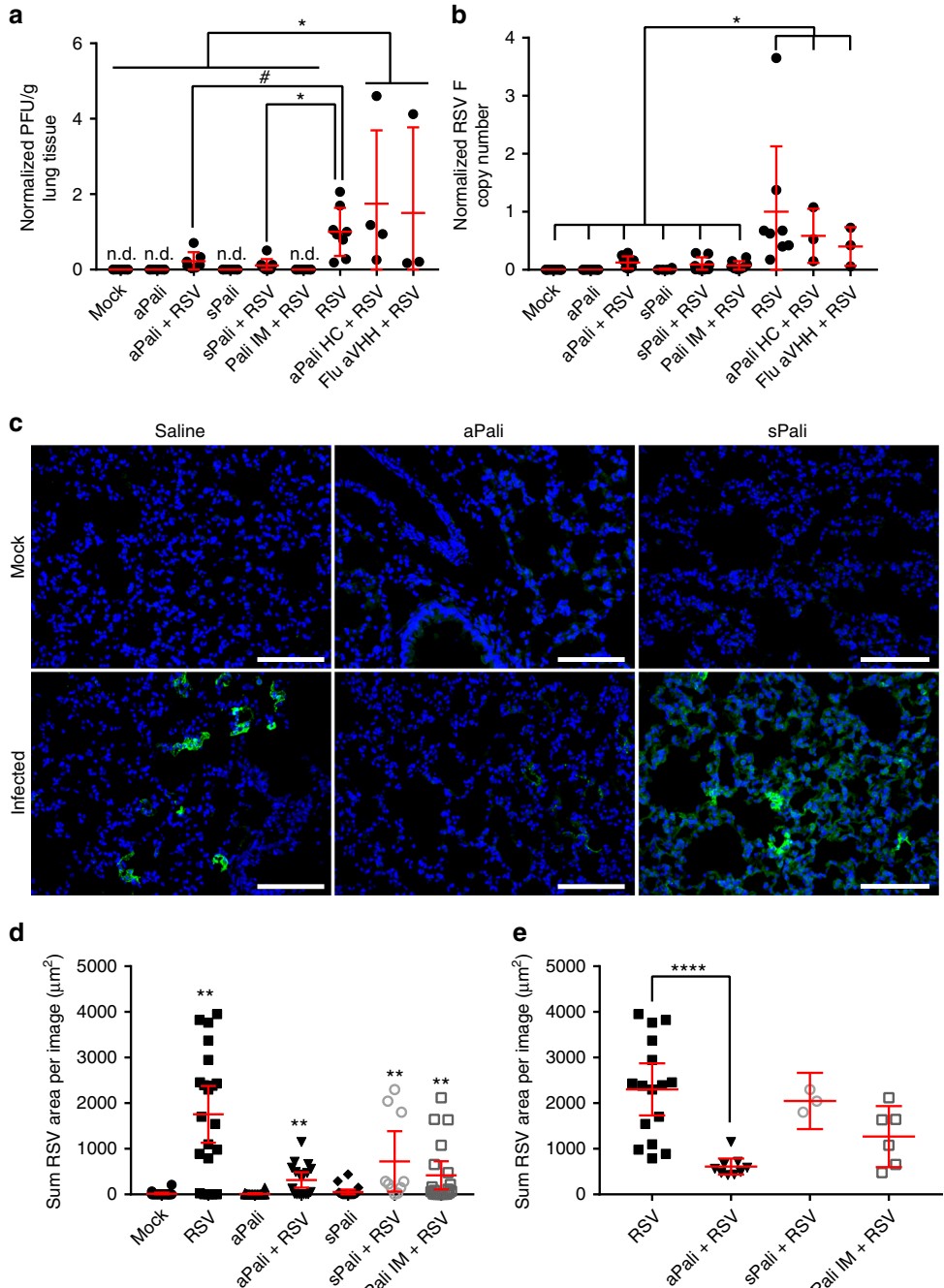

**Fig. 5** mRNA-expressed palivizumab prevents RSV infection in vivo. Mice lungs were transfected with 100 μg of naked aPali mRNA. Saline, negative, and irrelevant mRNA controls are indicated. 24 h later, mice were infected with RSV L19. At 4 days post infection, mice were sacrificed and lungs were excised. A total of 10 mice were used per group, out of which, 8 were processed for mRNA, protein, and virion extraction, with 2 being prepared for cryosectioning. aPali HC and Flu aVHH controls, along with a RSV only group, were performed in a separate experiment. **a** Virion content of mice lung homogenate supernatants was assayed by plaque assay. Error bars represent standard deviation. Asterisk indicates $p < 0.05$ (t-test, one-tailed). Hash indicates $p = 0.07$. **b** Mean RSV F copy number was determined by qRT-PCR. Data was normalized by saline control. Error bars represent standard deviation. Asterisk indicates $p < 0.05$ (t-test, one-tailed). **c** Cryosections were stained with panRSV antibody (green). Scale bar represents 100 μm. **d** Quantification of mean volume of RSV signal per frame from microscopy images in part **c**. Error bars represent 95% confidence intervals. Asterisks indicate $p < 0.01$ (Kruskal–Wallis, Dunn's multiple comparisons). **e** For thresholded comparison, images with less than 400 μm³ signal were excluded. Error bars represent 95% confidence intervals. Asterisks indicate $p < 0.0001$ (Kruskal–Wallis, Dunn's multiple comparisons)

(Supplementary Fig. 12). In the RSV aVHH mRNA treated animals, but not the aPali or sPali mRNA treated animals, we observed an increase only in the CCL3 and CCL5 concentrations. Overall, we have demonstrated that mRNA-expressed neutralizing antibodies in the lung are an effective method of preventing RSV infection in vivo without adverse cytokine responses.

**Adapting the GPI anchor to single-chain antibodies**. To demonstrate that the GPI anchor is adaptable to other neutralizing antibodies, we designed and produced an mRNA encoding for a single-domain camelid antibody (VHH), capable of neutralizing RSV at much lower antibody titers, linked to the DAF GPI membrane-anchor sequence, termed RSV aVHH[27].

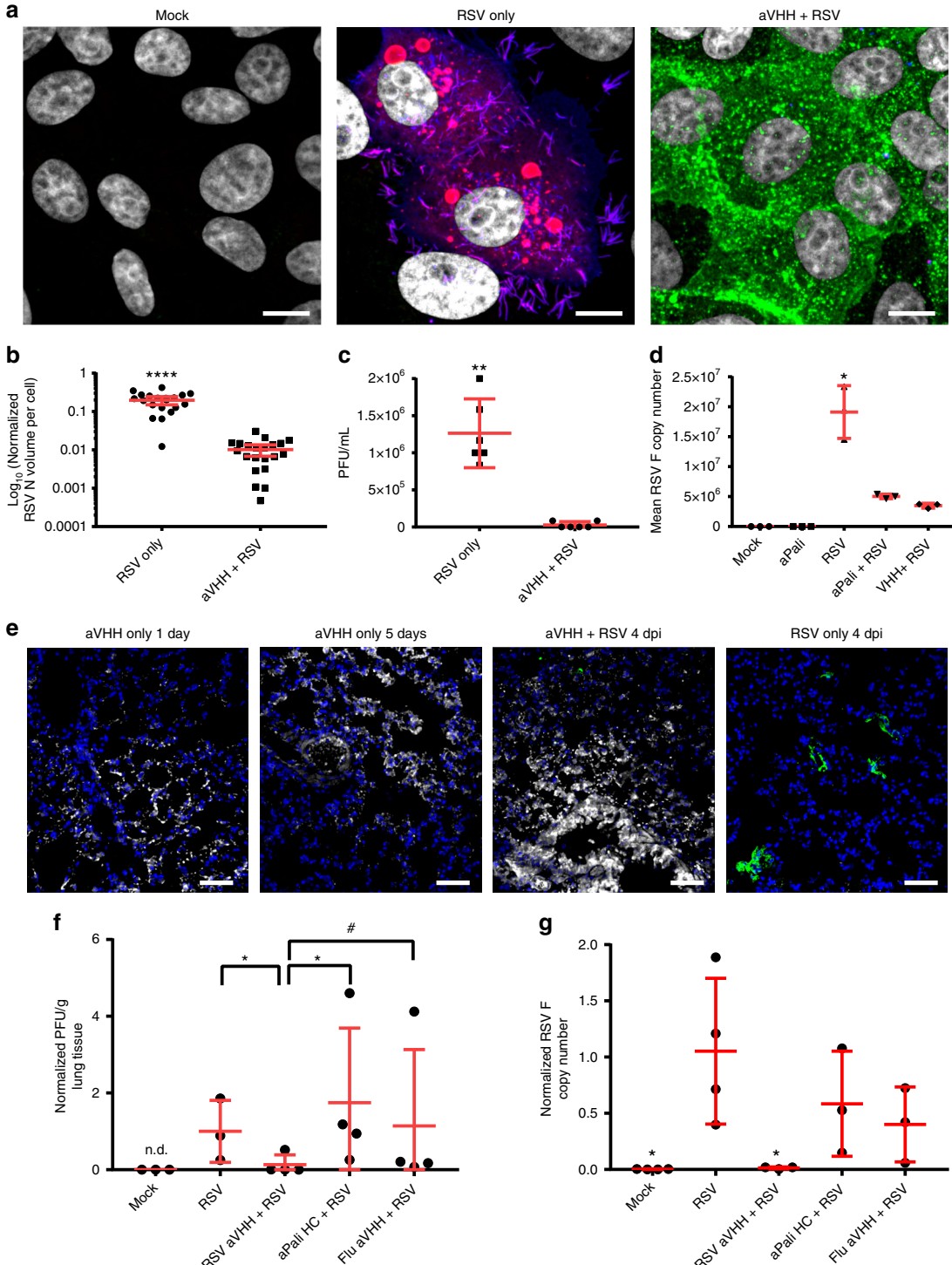

Single-chain antibodies offer several advantages over whole IgG for mRNA delivery, including: (1) they are much smaller than whole IgG, so more molecules are delivered per gram of mRNA (2) shorter mRNAs are translated more efficiently due to increased ribosome density, and (3) single-chain antibodies are expressed by one mRNA and thus do not require heavy and light chain association to actively bind antigen[32]. Like aPali, the expressed RSV aVHH was primarily localized to the plasma membrane in vitro and protected cells from RSV infection (Fig. 6a), as demonstrated by the quantification of RSV N volume in either untransfected or transfected cells (Fig. 6b). Importantly,

the RSV N signal observed in cells transfected with RSV aVHH was derived only from the initial virus inoculum; no detectable replication occurred in the transfected cells. To demonstrate that this effect was neither cell-type specific, nor a consequence of mRNA delivery alone, we transfected A549 cells with RSV aVHH mRNA or a membrane-anchored influenza-neutralizing camelid antibody, Flu aVHH mRNA (Supplementary Fig. 13). RSV infection was only abrogated in cells transfected with RSV aVHH mRNA, demonstrating that transfection with any mRNA is insufficient to prevent RSV infection. Delivery of RSV aVHH mRNA drastically reduced RSV titer, with 3 of the 6 replicates

**Fig. 6** Adapting the GPI anchor to single-chain antibodies. **a** Vero cells were transfected with vehicle control or 1 μg of either RSV or Flu aVHH mRNA. After overnight incubation, cells were infected for 24 h before being fixed and stained for RSV N (red), panRSV (blue), and VHH (green). Scale bar represents 10 μm. **b** Quantification of mean volume of RSV N signal per cell from microscopy images in part (**a**). Error bars represent 95% confidence intervals. Asterisks indicate $p < 0.0001$ (Mann–Whitney U test). **c** Cells were transfected with vehicle control or 1 μg of RSV aVHH mRNA and infected with RSV at MOI of 0.1 before being analyzed by plaque assay. Error bars represent standard deviation. Asterisks indicate $p < 0.01$ (Mann–Whitney U test). **d** Cells were transfected with vehicle control or 1 μg of either aPali or RSV aVHH mRNA. Cells were infected with RSV for 24 h before RNA was isolated. Mean RSV F copy number was determined by qRT-PCR and normalized to the mock control group. Asterisk indicates $p < 0.05$ (Kruskal–Wallis test, Dunn's multiple comparisons test compared to mock). **e** Mice lungs were transfected with saline or 100 μg of either naked RSV aVHH, aPali HC only, or Flu aVHH mRNA. aPali HC, and Flu aVHH controls, along with a RSV only group, were performed in a separate experiment – these data are repeated from Fig. 5a, b. Uninfected mice were sacrificed on days 1 and 5 post transfection. 1 day after transfection, mice were infected intranasally with RSV L19. Lungs were cryosectioned and stained for VHH (white) and panRSV (green). Scale bar represents 50 μm. **f** Infected mice lung homogenate supernatants from part (**e**) were analyzed by plaque assay. Error bars represent standard deviation. Asterisk indicates $p < 0.05$ (t-test, one-tailed). Hash indicates $p = 0.07$. **g** Infected mice lung homogenates from part (**e**) were analyzed by qRT-PCR for mean RSV F copy number. Data was normalized by RSV only control. Error bars represent standard deviation. Asterisk indicates $p < 0.05$ when compared to RSV (t-test, two-tailed)

producing no detectable plaques (Fig. 6c). We also evaluated the RSV titer in response to delivery of RSV sVHH or Flu sVHH mRNA, finding that while RSV sVHH completely prevented plaque formation, Flu sVHH had no effect on viral titer (Supplementary Fig. 14). Additionally, RSV aVHH mRNA transfection significantly reduced RSV F copies in infected cells, by 81.8%, compared to untransfected controls (Fig. 6d). These data indicate that delivery of RSV aVHH mRNA potently inhibited RSV infections in transfected cells.

Finally, to demonstrate that RSV aVHH mRNA is effective as a prophylactic in vivo, we delivered 100 μg of RSV aVHH mRNA to mice lungs. Mice were treated with 100 μg of either Flu aVHH or aPali heavy chain mRNA only as an irrelevant mRNA control. One day later, we infected the animals with RSV L19. At 4 dpi, animals were sacrificed, and their lungs were assessed by microscopy, qRT-PCR, and plaque assay. First, we found that RSV aVHH was readily detectable by immunofluorescence in transfected lungs, increasing in brightness from 1 day to 5 days post transfection (Fig. 6e). Additionally, RSV aVHH mRNA delivery reduced RSV staining compared to untransfected controls at 4 dpi. RSV titer in infected mice was significantly higher than both RSV aVHH treated and mock animals, with two of the RSV aVHH treated mice having no detectable plaques (Fig. 6f). Mock animals and mice treated with RSV aVHH mRNA had significantly lower RSV F copies than RSV only animals and mice treated with either aPali heavy chain only or Flu aVHH irrelevant mRNAs (Fig. 6g). Overall, these data demonstrate that (1) the GPI anchor sequence can be used interchangeably between IgG and single-chain antibodies, and (2) mRNA-expressed RSV aVHH is highly effective at preventing RSV infection both in vitro and in vivo.

**Anchored anti-RSV VHH protects from delayed RSV infection**. To demonstrate that the anchor enhances the half-life of the small camelid antibodies in vivo, we delivered 100 μg of either: aPali, sPali, RSV sVHH, RSV aVHH, or Flu aVHH (as an irrelevant mRNA control) mRNA to mice lungs. 7 days post transfection, we infected the animals with RSV L19, and at 4 dpi, animals were sacrificed, and their lungs were prepared for plaque assay and qRT-PCR (Fig. 7a, b). First, we observed that sPali, aPali, RSV sVHH, and RSV aVHH mRNAs prevented significant RSV titer when compared to either the RSV only group or control animals treated with Flu aVHH mRNA. We also found that sPali and RSV aVHH significantly reduced RSV F copies compared to both RSV and Flu aVHH treated animals. Animals treated with aPali mRNA had reduced RSV F copies compared to Flu aVHH treated animals. It is important to note here that although aPali has a significance level of only $p = 0.06$ (t-test) compared to Flu-aVHH in qPCR, it is significantly different at the level of the plaque

assay, which is a biologically relevant assay in terms of viral pathogenesis and disease. These data suggest that while the half-lives of whole antibodies, such as sPali and aPali, are sufficient to provide prolonged protection from RSV infection, the addition of the GPI membrane anchor is necessary to increase the half-life of the single chain anti-RSV VHH antibodies, leading to a comparable inhibition of RSV. Finally, we observed the prolonged accumulation of RSV aVHH in the lungs of mice sacrificed 28 days after mRNA delivery (Fig. 7c). We detected strong staining throughout the lung tissue, with significant RSV aVHH signal along the apical epithelial surface, suggesting that RSV aVHH is present in the lung for extended periods of time.

## Discussion

Here, we show mRNA-based expression of neutralizing antibodies in the lung to prevent RSV infection in vivo. We demonstrated that GPI-anchored mRNA-expressed antibodies are retained on the plasma membrane of transfected cells and that expressed neutralizing antibodies with and without a membrane anchor can prevent infections in vitro and in vivo, in an RSV model system. We determined that aPali inhibited RSV by preventing fusion of viral particles with the membrane of transfected cells. In addition, we found that most of the mRNA-expressed neutralizing antibodies we tested did not alter baseline cytokine levels, an important safety concern as undesirable inflammation could limit the translational application of this approach. We observed increases in the CCL5 and CCL3 concentrations in the RSV aVHH mRNA group without adverse effects in lung immunostaining. We also showed that the GPI anchor is adaptable to other neutralizing antibodies, such as highly-potent VHHs, where lung retention was greatly enhanced compared with historical half-life data for VHH antibodies. Theoretically, since the delivery characteristics are based on the mRNA properties, many expressed therapeutic proteins could be anchored in such a manner.

Interestingly, we found that expressing antibodies with the GPI anchor resulted in far less accumulation of the antibody in the ER and secretory membrane system compared to the secreted antibody. As we only evaluated the DAF GPI linker, we can only speculate if the improved trafficking is linker specific, but it is clear that the linker effected the localization. It is possible that the linker is improving antibody assembly, as the heavy chains may be less mobile in the lumen of the ER, promoting heavy and light chain interactions, and may improve trafficking to the plasma membrane[25]. Future work will measure linker effects on the efficiency of antibody assembly when expressed from mRNA. Linkers with higher cleavage rates could be incorporated to more reliably liberate the antibody from the cell in cases where secretion is critical.

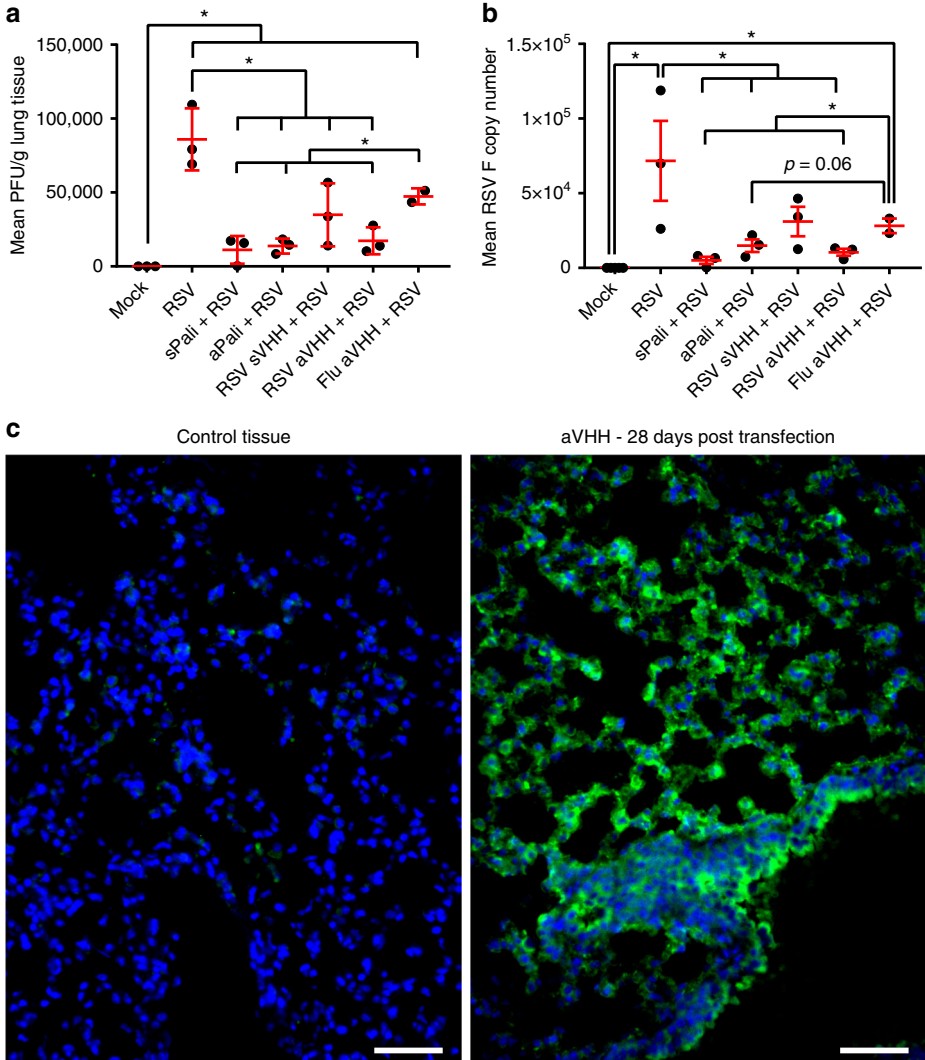

**Fig. 7** Anchored anti-RSV VHH protects from delayed RSV infection. Mice lungs were transfected with 100 µg of naked aPali, sPali, RSV sVHH, RSV aVHH, or Flu aVHH mRNA. RSV and negative controls are indicated. 7 days later, mice were infected with RSV L19. At 4 days post infection, mice were sacrificed and lungs were excised. **a** Virion content of mice lung homogenate supernatants was assayed by plaque assay. Error bars represent standard deviation. Asterisk indicates $p < 0.05$ ($t$-test, one tailed). **b** Mean RSV F copy number was determined by qRT-PCR. Error bars represent standard deviation Asterisk indicates $p < 0.05$ ($t$-test, two tailed). **c** Mice lungs were transfected with 100 µg of naked RSV aVHH mRNA. Control tissue was transfected with saline only. Mice lungs were harvested 28 days later and prepared for cryosectioning and were stained for VHH (green). Scale bar represents 50 µm

Additionally, our mRNA imaging data suggested that approximately half of the naked mRNA, when delivered by aerosol to the lung, was free of the endosomal compartment, in contrast to typical nanoparticle delivery where only a small fraction of RNA is delivered to the cytosol[33,34]. This clearly indicates that further research should focus on how droplet size, velocity, and buffers are facilitating cytosolic delivery. We did find though, that Viromer RED delivery of aPali mRNA was equally as effective at preventing RSV infection as naked mRNA delivery at 40 µg, and that in vivo-jetPEI exacerbated cytokine responses without significantly reducing viral titers, indicating the importance of the delivery approach on outcomes and safety. Moreover, we found that delivery of mRNA diluted in water, without infection, resulted in no significant inflammatory response. This result is critical, especially in the lung, where inflammation can have severe consequences and because a significant upregulation in cytokines could inhibit prophylactic protein expression.

While previous studies have used GPI-anchored antibodies for high-throughput screening purposes and HIV inhibition in vitro, we report the use of antibodies anchored to the surface of cells as

a prophylactic approach for lung infections[35,36]. Importantly, expressing anchored neutralizing antibodies on the surface of lung cells provides several benefits over systemic delivery by: (1) preventing the neutralizing antibody from diffusing away from the tissues infected by viruses, (2) raising the local concentration of antibody in the target organ, and (3) are likely increasing the persistence of antibodies on the mucosal surface. Together, these benefits point to a prophylactic approach that could be more efficacious than systemic neutralizing antibody delivery, while avoiding systemic delivery. Since children receive 15 mg/kg of palivizumab during treatment, this could significantly reduce costs and mitigate the risk of anti-human antibody responses. Critically, delivery of 100 µg of mRNA expressing either the secreted or anchored palivizumab was as protective as a therapeutic intramuscular dose of the commercially available palivizumab antibody.

These properties are more profound for single-chain antibodies. Single-chain antibodies are smaller than whole immunoglobulins, have larger diffusion rate constants, and are cleared much more rapidly, with serum half-lives ranging from 30 min to

2 h[37–39]. Anchoring these small, high-affinity, neutralizing antibodies to the plasma membrane overcame these issues while allowing for more mRNA copies to be delivered per equivalent mass. We found that RSV aVHH mRNA delivery was highly effective at mitigating RSV infection both in vitro and in vivo. Importantly, RSV aVHH mRNA delivery and infection 1 day later with RSV resulted in statistically similar data as mock infected animals by both plaque assay and qRT-PCR. Critically, when delivered 7 days prior to infection, RSV aVHH, but not the secreted form, prevented RSV infection as effectively as sPali. We also detected robust expression of RSV aVHH in the lung 28 days after infection. These results support our hypothesis that membrane anchored neutralizing antibodies can be retained in the lung. Moreover, while therapeutic use of VHH antibodies has been limited due to their rapid clearance from the body, the results presented here indicate that mRNA-expressed membrane-anchored neutralizing camelid antibodies may represent a suitable alternative.

Additionally, in cases where the Fc region of the antibody enhances neutralization of the pathogen, such as HIV, anthrax, and influenza, a sequence encoding the Fc could be inserted between the single-chain antibody coding region and the GPI anchor sequence[40–43]. Because they do not rely on heavy and light chain association, sequences encoding single-chain antibodies also offer the ability to multiplex several anchored neutralizing antibodies in a single cocktail, either for targeting multiple pathogens or strains.

Here we used RSV as a model system to test the prophylactic benefits of synthetic mRNA-expressed neutralizing antibodies. However, it is important to stress that mRNA encoding broadly neutralizing antibodies that target other viruses can be used to prevent infection, such as influenza or human metapneumovirus[44]. Additionally, the membrane anchor can be easily adapted to other neutralizing antibodies or proteins in cases where systemic antibody therapy is undesirable. Moreover, we emphasize the use of the anchor on single chain antibody constructs, which will allow for multivalent prophylaxis against additional pathogens and strains. Overall, we feel our data suggests that pulmonary expression of neutralizing antibodies has the potential to be a new prophylaxis approach to protect the lung from infections.

## Methods

**Cell lines and virus culture**. Vero cells (African green monkey kidney cells, American Type Culture Collection CCL-81), A549 cells (human lung epithelial cells, ATCC CCL-185), or HEp-2 (human epithelial cells, ATCC CCL-23) were cultured in DMEM (Lonza) supplemented with 10% fetal bovine serum (FBS) (Hyclone) and 100 U mL$^{-1}$ penicillin and 100 mg mL$^{-1}$ streptomycin (Life Technologies). Cells were plated on No. 1.5 coverslips (Electron Microscopy Sciences) 1 day prior to infection. Both Vero and HEp-2 cell lines were authenticated by ATCC and were checked for Mycoplasma contamination in our laboratory. HEp-2 cells are commonly used to propagate RSV in culture.

Human RSV A2 (ATCC VR-1544) and human RSV L19 (a gift from Martin L. Moore's lab, Emory University) was propagated in HEp-2 cells when the cells were >80% confluent. The media was removed and cells were washed with DPBS (without Ca$^{2+}$ and Mg$^{2+}$, Lonza), and virus was added at a multiplicity of infection (MOI) of 0.1 for 1 h before adding complete medium to the inoculum. Cell-associated virus was harvested by scraping the cells when about 90% of cytopathic effects was visualized (about 96 hpi). Virus was then vortexed briefly, aliquoted, and stored at −80 °C. For L19 virus, the cell suspensions were sonicated at 30% amplitude, with 1 s on/off pulses and centrifuged at 800×g at 4 °C. Supernatants were pooled and 1 mL aliquots were snap frozen in liquid nitrogen and stored at −80 °C. Virus titers were measured via plaque assay.

**Antibodies**. For immunostaining, primary antibodies used were mouse anti-RSV N (Abcam, Cat. No. ab22501), mouse anti-RSV G (Abcam, Cat. No. ab94966), rabbit anti-camelid VHH (GenScript, Cat. No. A01860-200), goat anti-panRSV polyclonal (Millipore, Cat. No. AB1128), rat anti-CD63 (BD, Cat. No. 564221), rabbit anti-Rab5 (Affinity BioReagents, Cat. No. PA3-915), and rabbit anti-Rab7 (Abcam, Cat. No. 137029). All primary antibodies for immunostaining

experiments were used at 1 μg/mL. Secondary antibodies used were donkey anti-human Alexa Fluor 647 (Jackson ImmunoResearch), donkey anti-goat Alexa Fluor 546, donkey anti-rabbit Alexa Fluor 488, and donkey anti-mouse Alexa Fluor 488 (all from Life Technologies). All secondary antibodies for immunostaining experiments were used at 4 μg/mL. For dSTORM experiments, secondary antibodies were donkey anti-mouse Alexa Fluor 647 (Life Technologies) and goat anti-human CF568 (Biotium), and both were used at 1 μg/mL.

For western blot, the primary antibodies used were a rabbit monoclonal anti phosphorylated eIF2α (Cell Signaling Technologies, Cat. No. 9721) and goat anti-pan-eIF2α (R&D Systems, Cat. No. AF3997) diluted 1:1000 in 5% BSA in TBS with 0.1% Tween-20. The secondary antibodies were a donkey anti-mouse IRDye 680RD (LI-COR) and a donkey anti-rabbit IRDye 800 (LI-COR) and were diluted 1:5000 in 5% BSA in TBS with 0.1% Tween-20.

**Construct preparation for in vitro transcription**. Plasmids for IVT were designed using the full length nucleotide sequences of palivizumab heavy and light chain. The coding region was preceded by 5′ UTR with Kozak sequence and followed by a 3′ untranslated region (UTR) derived from the mouse alpha globin sequence. Sequences were human codon optimized and inserted in a pMA-7 vector (Thermo Fisher Scientific, GeneArt) to be used as a template for mRNA synthesis. For the anchored version of palivizumab, a GPI linker based on the human decay accelerating factor (DAF) was appended to the Fc region of the heavy chain and synthesized in the same manner as the secretory version[45].

V5 is a 14 amino acid epitope tag that was inserted into the 3′ terminus of the light chain construct through two fragment Gibson assembly. The linear backbone was PCR amplified from the insertion site in the forward and reverse directions (primers 1 and 2, Supplementary Table 1). The insert was 104 bp in size–42 bases for the V5 tag and 31 × 2 bases for the overlapping segments that correspond to the aPali light chain backbone. This insert was synthesized via PCR (primers 3 and 4, Supplementary Table 1). All PCR products were 0.8% agarose gel purified.

The backbone was assembled with 3× molar excess of insert in 5 μL total volume of NEBuilder® HiFi DNA Assembly Master Mix, according to NEB's protocol. After incubation for 1 h at 50 °C, 1 μL of reaction mixture was used to transform NEB 5-alpha competent E. Coli. Ampicillin plates were used to select 10 colonies and outgrowth performed at 30 °C for 24 h. Subsequent plasmids from each colony were Sanger sequenced to ensure incorporation of the V5 tag at the 3′ terminus of the light chain. Immunofluorescence staining confirmed that V5 tag insertion did not negatively impact heavy and light chain association or ability to knock down RSV infection.

The anchored and secreted anti-RSV VHH sequence, F-VHH-4 from Rossey et al., or anchored anti-Flu VHH from Gaiotto and Hufton, flanked by UTRs, was ordered as a DNA geneblock from IDT[27,46]. The geneblock was PCR amplified with primers that contained overlapping regions for the PMA-7 vector (primers 5 and 6, Supplementary Table 1). This VHH PCR product was cloned into the PCR amplified pMA7 vector (primers 7 and 8, Supplementary Table 1) through Gibson assembly using NEB Builder with 3× molar excess of VHH insert, as described above. All reaction transcripts were 0.8% agraose gel purified prior to assembly reaction. Subsequent plasmids from each colony were Sanger sequenced to ensure desired sequence fidelity.

**Synthetic mRNA in vitro transcription**. Plasmids were linearized with Not-I HF (New England Biolabs) overnight and PCR purified using PCR clean-up kit (Qiagen), prior to in vitro transcription (IVT) using a T7 mScript kit (Cellscript) following the manufacturer's instructions. Equimolar ratios of ATP, GTP, and CTP were used alongside N1-methylpseudouridine-5'-triphosphate (TriLink). RNAs were capped using 2'-O-Methytransferase followed by enzymatic addition of a poly-A tail, both according to the mScript kit instructions. The capped and tailed mRNAs were then purified using an RNeasy kit (Qiagen) and treated with Antarctic Phosphatase (New England Biolabs) for 1 h (New England Biolaboratories), and purified again. RNA concentration of the purified mRNA was measured the RNA was stored at −80 °C until further use.

**Transfections in vitro**. Vero or A549 cells were transfected using either Viromer Red (Lipocalyx) or Neon electroporation system (Invitrogen), according to the manufacturer's instructions, into a 24 well plate (for imaging and plaque assays) and were transfected with the indicated amount of mRNA per 200,000 cells. For mRNA encoding whole IgG, heavy chain and light chain mRNAs were combined in a 4:1 mass ratio for equimolar conditions. 20 h post-transfection, cells were fixed and immunostained with or without permeabilization.

**Immunostaining**. Vero or A549 cells were fixed with 4% paraformaldehyde (PFA) (Electron Microscopy Sciences) for 10 min at room temperature before permeabilization with 0.2% Triton X-100 (Sigma) for 5 min at room temperature. Then, cells were blocked by incubation with 5% bovine serum albumin (Calbiochem) for 30 min at 37 °C before being incubated with primary antibody for 30 min at 37 °C. Cells were then washed with PBS and incubated with secondary antibody for 30 min at 37 °C. Multiple antibody labeling was performed simultaneously after checking cross-reactivity. Nuclei were then stained with 4',6-diamidino-2-

phenylindole (DAPI) (Life Technologies), and coverslips were mounted onto glass slides with Prolong Gold (Life Technologies).

**Plaque assay**. For viral titer in vitro, Vero cells were transfected with or without aPali, RSV sVHH, or RSV aVHH mRNA, using the Neon system, into a 24 well plate and incubated until fully confluent. Flu-aVHH mRNA was used as a negative control. Either 50 µL of serially diluted lung supernatant (for in vivo studies) or 50 plaque forming units (PFU) or multiplicity of infection (MOI) of 0.01, 0.1 or 1 (for direct measurements or supernatant measurements, respectively) of RSV A2 stock was added to the cells. Cells were incubated at 37 °C for 1 h with shaking every 15 min. At 1 hpi, 1 mL of overlay media, consisting of either 1.2% Avicel (FMC Biopolymer) or 1.2% methylcellulose (Sigma-Aldrich) in 1× DMEM with 2% FBS and 100 U mL$^{-1}$ penicillin and 100 mg mL$^{-1}$ streptomycin (Life Technologies), was added. At either 6 dpi (RSV for A2) or 8 dpi (for RSV L19), cells were washed and fixed with 4% paraformaldehyde, blocked with 5% BSA, and stained with a goat anti-panRSV polyclonal and HRP-conjugated donkey anti-goat secondary antibody. A precipitating peroxidase substrate (TrueBlue, KPL) was then added and cells were incubated at room temperature for 10 min before a final wash with water. Plaque count was converted using the dilution factor to generate viral titer in PFU/mL and, in the case of in vivo assays, lung titers were normalized by lung weights. In order to make comparisons between animals assayed with different viral stocks, viral titers were normalized by the mean of the RSV only group for a given experiment

**Animal studies**. Six-week-old to 8-week-old female BALB/c mice (Charles River Laboratories) were maintained under pathogen-free conditions in individually ventilated and watered cages kept at negative pressure. Food was provided to mice ad libitum. Animals were acclimatized for at least 6 days before the beginning of experiments. Animals were randomly distributed among experimental groups. Researchers were blinded to animal group allocation during data acquisition. Animals were sacrificed by CO$_2$ asphyxiation. Infected animals were handled and kept under BSL-2 conditions until euthanized. All animals were cared for according to the Georgia Institute of Technology Physiological Research Laboratory policies and under ethical guidance from the university's Institutional Animal Care and Use Committee (IACUC) following National institutes of Health (NIH) guidelines. Number of animals used in each experiment is detailed in Supplementary Table 1.

**Palivizumab intramuscular injection**. Mice were anesthetized under constant isoflurane (Henry Schein Medical) with oxygen and kept on a heating pad. The *anterior tibialis* muscle was then shaved and injected with 15 mg/kg of palivizumab (either unlabeled or labeled with DyLight 680 per the manufacturer's protocol) in 25 µL using a 29 gauge needle. Near-IR imaging was performed using a Fluobeam (Fluoptics) on anesthetized mice. Mice were euthanized 24 h post injection and organs were imaged for palivizumab biodistribution in organs pre- and post-harvesting using the Fluobeam.

**Intratracheal aerosol mRNA transfections**. Mice were anesthetized using 87 mg/kg ketamine and 16 mg/kg xylazine. The trachea was visualized by inverting the mouse on a 45° tilting intubation stand (Hallowell EMC) using the upper teeth and illuminating the trachea with a flex light. A 1.22 mm endotracheal tube (ET, Hallowell EMC) was then inserted to intubate the animal. A MicroSprayer Model IA-1C connected to a High Pressure Syringe Model FMJ-250 (both from Penn-Century) was then inserted such that the tip of the sprayer nozzle was 1 mm past the tip of the ET tube. The MicroSprayer was then actuated using a custom Pump 11 Elite Nanomite handheld syringe pump (Harvard Apparatus) that allowed for repeatable pressure and volume delivery of 50 µL. To find the best delivery vehicle, we compared Viromer RED and in vivo-jetPEI against naked mRNA at 40 µg in nuclease free water. For naked mRNA transfections, the appropriate amount of mRNA was mixed with water to the desired concentration. For Viromer RED transfections, 0.2 µL of Viromer RED (Lipocalyx) per 1 µg of mRNA was mixed with Buffer Red. The mRNA, also diluted in Buffer Red, was then added to the Viromer solution, per the manufacturer's instructions. For in vivo-jetPEI transfections, 40 µg of mRNA was delivered with 6.4 µL of vehicle diluted in 5% glucose in water. In order to mitigate any possible double stranded RNA-dependent protein kinase (PKR) response, Imidazolo-oxindole PKR inhibitor C16 (Sigma-Aldrich) was added to all mRNA preparations to a final concentration of 375 µM. To find an optimum mRNA amount for delivery, we tested aPali at 20 µg and 100 µg without any delivery vehicle. To assess the immune response of mRNA, we delivered 100 µg of aPali, sPali, or RSV-aVHH mRNA without C16. We also tested the efficacy of aPali mRNA (100 µg) against RSV at day 1 and day 4 post infection.

**MTRIP labeling of mRNA**. We designed four 2' O-methyl RNA-DNA chimeric oligonucleotides, each containing a 17–18 nucleotide binding region and 5–7 poly (T) linker[28–30]. Oligos were complementary to 4 adjacent sequences across the mRNA 3' UTR regions and contained 3–4 amino-modified thymidines each, as well as a 5' biotin modification (Biosearch Technologies). Labeled oligos were synthesized by conjugating either Cy3B NHS ester (GE Healthcare) or DyLight 650 (Thermo Fisher) to the amine groups on the oligonucleotides followinhisg the manufacturer's protocol. Complete MTRIPs were assembled by incubating the

labeled oligos with Neutravidin (Pierce) for 1 h at RT followed by filtration using 30 kDa MWCO centrifugal filters (Millipore). mRNA was buffer exchanged into 1× PBS, heated to 70 °C for 10 min, and immediately placed on ice. Denatured mRNA was then combined with MTRIPs in a 1:1 ratio for each MTRIP and incubated at 37 °C overnight. Labeled mRNA was then buffer exchanged into water and concentrated using a 30 kDa filter before being injected into the animals.

**RSV infections in mice and tissue processing**. One day after transfection, mice were anesthetized using cotton soaked with isofluorane in a small chamber. After breathing rate had slowed sufficiently, mice were inoculated with $5 \times 10^5$ PFU of RSV L19 in 50 µL, with 25 µL in each nostril. Mice were weighed before infection and then once daily to monitor weight loss. Mice were sacrificed by CO$_2$ asphyxiation at 4 dpi. Lungs were harvested in cold DMEM without any supplements and were either stored on ice for imaging or snap-frozen in liquid nitrogen for protein and RNA extractions. Snap frozen tissues were thawed and homogenized using a Gentle MACS Tissue Dissociator (Miltenyi) and centrifuged to collect supernatants for plaque assay and cytokine analysis. Tissue pellets from this step were used for RNA extraction.

For plaque assay, neat and serially diluted lung homogenates were added to confluent Vero cells and processed as described above. Dilutions with individual plaques were counted and averaged across 4 replicate wells to generate viral titer in PFU/mL. Viral titer was then normalized by lung weight.

Lung tissues collected in DMEM were washed in PBS and fixed in 4% paraformaldehyde overnight. Tissues were then washed three times in PBS and were cryoprotected using 30% sucrose (Sigma) in 1× PBS until tissues sank to the bottom of the container. Tissues were then frozen in OCT compound (Tissue-Tek) and cryosectioned (10 µm sections). Tissue sections were then immunostained as described above.

**Flow cytometry**. Mice lungs were collected in DMEM and washed to remove any blood. The tissues were then dissociated using the Miltenyi Lung dissociation kit, following manufacturer's instructions with slight modifications[47,48]. Briefly, after coarse digestion of lungs, 0.1 mg/mL soybean trypsin inhibitor (Sigma) 60 U/mL DNAse I and 1 mg/mL type XI collagenase was added and tissues were digested at 37 °C for 30 min. Then, lungs were finely homogenized in Miltenyi Gentle MACs, followed by addition of 0.1 M EDTA in FBS to inactivate the enzymes. Cell suspension was then passed through 70 µm filter, washed with PBS by centrifugation at 800×$g$ for 10 min. RBCs were lysed using ACK lysis buffer (Alfa Aesar) and washed with DPBS. Cells were then counted and 1–2 million cells were stained with live/dead marker (Life Technologies) before being incubated with rabbit anti-V5 (1:250, Abcam ab9116) antibody followed by AlexaFluor 488 conjugated donkey anti-rabbit secondary antibody (2 µg/ml, Invitrogen A-21206). Cells were then fixed with BD Cytofix/Cytoperm (BD Biosciences) and acquired on a BD LSRFortessa flow cytometer. Compensation was performed using beads (BD Biosciences) labeled with fluorophore conjugated antibodies. Live cells were gated on total cells and analyzed for V5 positive cells. Data was analyzed using Flow Jo software (Treestar, Inc.).

**Western blot**. For positive control of phosphorylated eIF2α, HFF cells were treated with 500 µM sodium arsenite (Sigma) for 1 h at 37 °C. Protein was extracted from cells and lung tissues in RIPA buffer (Pierce) containing 1× complete protease and phosphatase inhibitor (Thermo Scientific) before being clarified by centrifugation. Protein concentration was calculated using a BCA assay (Pierce) following the manufacturer's protocol. Lysates were stored at −80 °C until further use.

For gel electrophoresis, 30 µg of lysates were mixed with 4× SDS loading buffer (LI-COR Biosciences), boiled for 10 min at 70 °C, chilled on ice, and loaded into wells of a Bolt 4–12% Bis-Tris precast gel (Life Technologies) alongside a molecular weight marker (LI-COR). Gel was run in an Mini Gel Tank system (Life Technologies) in 1× MOPS running buffer (Life Technologies) at a constant 200 V for 32 min. Protein was then transferred to 0.45 µm pore nitrocellulose membranes (Life Technologies) in 1× Bolt western transfer buffer (Life Technologies) at a constant 12 V for 1 h using an Mini gel blot module (Life Technologies).

Nonspecific binding to blot was blocked using 5% BSA in 1× TBS at room temperature. Primary antibody, including a mouse anti-GAPDH antibody (Genetex, Cat. No. GTX627408) loading control (diluted 1:1000), was then applied in blocking solution with 0.1% Tween-20 and allowed to incubate overnight at 4 °C. Blots were washed three times with 1× TBS containing 0.1% Tween-20 (PBST). Secondary antibody was then applied and allowed to incubate for 10 min before blots were again washed three times with PBST. Blots were imaged using an Odyssey CLx IR scanner (LI-COR). Only linear contrast enhancements were performed for the final representative images.

**qRT-PCR**. Total RNA was extracted from cells or tissues using an RNeasy kit (Qiagen) and cDNA was synthesized using 250 ng of total RNA and a High Capacity cDNA synthesis kit (Applied Biosystems) with random hexamers. Extent of RSV infection was assessed using an absolute quantification method based on RSV F gene copy numbers[49]. In order to perform absolute quantification of RSV F mRNA quantity using RSV F gene standards, we normalized the amount of RNA (250 ng) used for cDNA synthesis and used the same amount of cDNA

for qRT-PCR. Primers used are in Supplementary Table 2. For cytokine analysis, Taqman gene expression assays were used for IL-6, IFN-β, IFN-γ, and CCL5 (Thermo Fisher). In order to make comparisons between animals assayed with different viral stocks, RSV F copy numbers were normalized by the mean of the RSV only group for a given experiment.

**Cytokine response measurement**. Lung homogenates were assessed for cytokine response using a custom Millipore Milliplex assay following the manufacturer's instructions. Briefly, samples were diluted 1 fold in assay buffer and assayed using a BioPlex 2.0. Observed and interpolated cytokine concentrations were used and undetected cytokines were set as 0.1 pg/mL. To assess the effects of delivered mRNA, diluted in water without C16, at day 1, production of IL12p70, IL6, TNFα (all from BD), CCL5, CCL3 (both from R&D Systems), and IFNα (Affymetrix) were measured from lung homogenates using ELISA kits following manufacturer's instructions.

To test the effect of PKR activation due to mRNA delivered with and without C16, we tested mice lung homogenates for phosphorylation of eIF2α using a Pathscan phospho-eIF2α kit (Cell Signaling Technology) diluted 1:1 in dilution buffer according to the manufacturer's instructions.

**Microscopy**. All images were acquired with a Hamamatsu Flash 4.0 v2 sCMOS camera on a PerkinElmer UltraView spinning disk confocal microscope mounted to a Zeiss Axiovert 200 M body with either a 63× NA 1.4 plan-apochromat objective for individual cells or a 20× NA 0.8 plan-apochromat objective for tissue sections. All 63× images were acquired with Volocity (PerkinElmer) with Z-stacks taken in 0.2 μm increments. Stitched tissue images were acquired automatically using the motorized ASI PZ-2150 stage in Volocity using 25% overlap with automatic alignment and without background correction. Linear contrast enhancements were applied to images for clarity. Unless indicated, all images in a given experiment were enhanced the same. All image quantification and analysis was completed on unenhanced data.

**dSTORM**. Images were acquired using a Bruker Vutara 352 with a 63× NA 1.2 plan-apochromat water immersion objective lens kept at room temperature. Samples were immunostained as above, but without nuclear counterstain since the 405 laser is used for dSTORM excitation. Samples were imaged in a buffer to enhance fluorophore blinking which contains: 50 mM Tris-HCl (pH 8.0), 10 mM NaCl, 10% glucose, 20 mM cysteamine, 1% 2-Mercaptoethanol, 169 AU/mL glucose-oxidase, and 1404 AU/mL catalase. Imaging buffer was prepared fresh daily and replaced after approximately 2 h. All images were acquired and analyzed using the Vutara SRX software. All images were processed using the denoise option set to 0.2. For particle height quantification, a 1 μm square region of interest (ROI) around each individual particle (identified by N staining signal) was analyzed. To ensure quantification of RSV virions, particles were only included if the XY FWHM was greater than 50 nm and the Z FWHM was greater than 100 nm. The mean of the aPali signal and mean of the RSV N signal was subtracted to determine mean particle distance above the aPali signal.

**Statistical analysis**. Results were plotted and statistical analyses were performed using Prism 7 (GraphPad). Power analysis was performed to ensure adequate sample size for experiments. Hypothesis tests were chosen and performed as appropriate, indicated in the figure captions.

## Data availability
All data relevant to this study are available from the corresponding author upon request.

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

## Acknowledgements

The authors would like to acknowledge Dr. Martin L. Moore for RSV L19 and help with the propagation protocol. The authors would also like to acknowledge Sommer Durham from the Analysis and Cytometry Core and Andrew Shaw from the Biomolecular Analysis core, Petit Institute for Bioengineering and Bioscience at the Georgia Institute of Technology. This work was funded by DARPA grant W911NF-15–0609. This work was also funded through contributions from Children's Healthcare of Atlanta.

## Author contributions

D.V., J.L.K., and P.J.S. conceived the overall project. P.M.T. and D.V. performed the in vitro experiments. P.M.T., D.V., K.E.L., S.B., J.L.K., and S.S.B. performed the in vivo experiments. K.E.L., P.M.T., and D.V. designed the mRNA constructs. D.V. and P.M.T. performed the data analysis. All authors contributed to the manuscript.

## Additional information

**Competing interests:** P.M.T., D.V., J.L.K., K.E.L., and P.J.S. have applied for patents related to this study. The remaining authors declare no competing interests.

