## [Peer Review File · Nature Communications]

Reviewers' comments:

Reviewer #1 (Remarks to the Author):

The authors show that Vero cells transfected with in vitro transcribed mRNAs encoding the heavy and light chain of a GPI membrane anchored variant (through the heavy chain) of palivizumab, results in cell membrane-associated expression of this antibody. Cells that express aPali after transfection with the corresponding mRNAs, are very resistant to infection with RSV. dSTORM imaging of aPali mRNA transfected and RSV infected Vero cells, shows that RSV virions are trapped at the cell membrane by the expressed aPali. A mixture of mRNAs encoding heavy and light chain aPali that was intratracheally delivered in mouse lungs by aerosol, gave rise to aPali expression at 4 h after administration. Intratracheal administration of a variant mRNA mix with a V5-tagged aPali light chain resulted in detectable light chain expression in 2 out of 3 mice. RSV challenged aPali mRNA "transfected" mice had an approximately 5-10 fold reduced RSV titer on day 4 after infection. Transfection of mRNA coding for a GPI-anchored variant of an RSV neutralizing single domain antibody resulted in detectable aVHH 5 days after transfection and suppressed RSV infection in vitro and in vivo.

The idea to deliver anti-microbial antibodies as in vitro transcribed mRNAs that encode these antibodies and in particular the GPI-anchor approach is very interesting. The presented micrographs are of high quality. Finally, it is surprising that naked mRNA distributed intratracheally as an aerosol, somehow ends up in the respiratory epithelial cells and gives rise to expression of the encoded antibodies.

Major remarks,

1. It is very unfortunate that results with negative control RNAs (e.g. encoding a reporter gene or better an irrelevant antibody or VHH) are missing. This is an important caveat of the study, in particular because synthetic mRNA, even when care is taken to use hypo-inflammatory mRNA, can induce type I and III IFN, which are cytokines with a strong antiviral effect. Negative control mRNA settings should be included in all experiments.
2. In vivo mRNA treatments were all performed in the presence of a chemical inhibitor of PKR. The authors should demonstrate that the inhibitor was necessary and has worked, for example by documenting eIF2 α or PKR phosphorylation after in vivo mRNA transfection in the absence of C16. In addition, the intratracheal mRNA delivery experiments could be performed in PKR knock out mice where the method should work without the inhibitor.
3. The expression level of a number of cytokines was determined in mouse lungs sampled from animals that had been treated with in vitro transcribed mRNA, subsequently infected with RSV and sacrificed on day 4 after infection. It is important to evaluate cytokine expression in lung samples at 8 or 24 hours after mRNA transfection, prior to RSV challenge. This could support the statement that the mRNA is poorly inflammatory in vivo.
4. There is only a minor difference in RSV neutralization activity between Pali and aPali mRNA. It would be very interesting to make a comparison of the RSV inhibitory activity of VHH with aVHH mRNA in vivo where the advantage of anchoring over the rapidly diffusing non-anchored VHH would perhaps be more pronounced.
5. An important potential advantage of the authors' approach is that mRNA delivery of an anchored antibody could result in more sustained protection against RSV challenge. It would

therefore be interesting to test in vitro and in vivo how long after transfection of aPali/aVHH compared with sPali/sVHH protection lasts. Thus start infections at 2 days or longer after mRNA administration.

6. The number of animals that were used is low and does not allow to conclude that there are significant differences between the groups based on statistical power. E.g. in Figure 6g, there appears to be no significant difference between RSV and aVHH + RSV. It is highly recommended to document the protection based on repeat experiments.

7. The PFU graph in figure 4b indicates a 4-8 fold difference in the number of plaques/g lung between the saline and jet PEI vs the VR and naked setup. The tissue slides in figure 4d suggest that the naked mRNA setting had no detectable RSV foci. Please explain this difference.

8. The right hand panel of figure 1c and the top right hand panel of figure 1d essentially show the same: Vero cells transfected with aPali mRNA and 24h later permeabilized and stained with an anti-human antibody. Yet the outcome looks very differently. Please explain this apparent discrepancy.

9. In figure 3e, lung cells from 1 out of 3 "transfected" mice did not score V5-positive. Does this reflect variability in the in vivo efficiency of the mRNA delivery? Please explain.

10. Results: ... a significant fraction of it was likely free in the cytosol. Why was this mRNA not detected (cfr. Figure 3b)?

11. It is unclear how the RT-qPCR for RSV F RNA was normalized. Please explain in the methods section.

12. The authors should document that the mRNA derived sPali is as potent (anti-RSV activity) as the licensed recombinant humanized antibody counterpart i.e. Palivizumab.

13. There is a risk that aPali produced in limited amounts on a mRNA transfected cell surface can enhance RSV virion attachment and subsequent entry. The authors should test an in vitro dose range of mRNA to assess this possibility.

14. The authors should test in a human cell line, e.g. A549 (human lung epithelial-derived cell line), that aPali can also be prevent RSV infection.

Minor comments:

1. X-axis labeling in 6d should be aVHH instead of VHH to concord with the description in the legend.

2. Last paragraph of the results: ... we delivered ... of aVHH or aPali mRNA by ... In figure 6e-f only aPali is shown.

3. Last paragraph of the introduction. ... expressed anchored neutralizing antibodies,... (3) were minimally immunogenic. "Immunogenic" is most often used to denote that something induces B or T cell responses. Such responses were not studied in the manuscript. Presumably the authors mean minimally inflammatory. However, it is not clear if this is the case cfr. remarks 2 and 3 above.

4. Methods, plaque assay: "mRNA knockdown" is commonly used for si- or shRNA mediated reduction of cellular mRNA.

5. Results, paragraph "Optimizing mRNA...": "To compare the therapeutic..." A therapeutic experiment, meaning treatment of already RSV-infected animals, is not described. Please adapt the statement.

Reviewer #2 (Remarks to the Author):

The paper explores potential of mRNA for delivery of antibodies against respiratory syncytial virus (RSV) and influenza. Addition of membrane anchoring sequence to antibody mRNA sequence helps to retain the antibody on the cell surface and improves cell protection against RSV. In vitro experiments are supported with in vivo studies demonstrating animal protection from the virus after the IgG delivery using naked modified mRNA delivered via intratracheal aerosol. Finally, the authors demonstrate that anchored sequence can be incorporated into other antibody increasing their efficacy as well.

Advantages:

1. The study explores an original idea of antibody expression on the surface of lung epithelial cells instead of secreting the antibody into intracellular space.
2. The paper expands our knowledge about antibody mRNA expression, transport and assembly inside the cell.

Disadvantages:

1. Authors expressed a humanized antibody in healthy mice, which can activate foreign body immune response and subsequently be a reason of resistance to the virus. Thus, a control with non-specific humanized antibody mRNA (or at least a scrambled mRNA) is needed.
2. Fig 4. 1) Standard deviations for certain groups are big; 2) We would expect to see standard deviation of similar size for different groups in the same experiment unless there is a certain reason for observation of an increased variance in certain groups. I suggest, that the experiments should be repeated with a higher animal number per group.
3. In the paper authors study immune response 5 days after mRNA injection. It has been shown in several studies (Mol. Ther. 20 (2012) 948-953; Biomaterials 109 (2016) 78-87) that immune response is usually gone 72h after mRNA injection, thus I suggest to check immune response markers after 6-24h.
4. It would be helpful if authors describe explicitly what they mean by Mock, Saline 1 and Saline 2 (Fig.4); add names of the cell lines to the descriptions for fig 1; show a picture of Vero cells infected with the virus uptaken by the cell instead of the bad resolution picture.

Response to reviewers' questions and comments: We have addressed all of the concerns posed by the reviewers as follows:

Reviewer 1:

1. *It is very unfortunate that results with negative control RNAs (e.g. encoding a reporter gene or better an irrelevant antibody or VHH) are missing. This is an important caveat of the study, in particular because synthetic mRNA, even when care is taken to use hypo-inflammatory mRNA, can induce type I and III IFN, which are cytokines with a strong antiviral effect. Negative control mRNA settings should be included in all experiments.*

Response to reviewer: We designed an mRNA encoding a GPI-anchored VHH antibody that neutralizes influenza virus (from Reference 46), termed Flu aVHH. We tested Flu aVHH *in vitro*, using microscopy (Figure 6A and B and Supplemental Figure 12) and plaque assay (Supplemental Figure 13), and *in vivo*, using qRT-PCR (Figure 7a) and plaque assay (Figure 7b). The Flu aVHH did not have significant effects on RSV *in vitro* and *in vivo*.

2. *In vivo mRNA treatments were all performed in the presence of a chemical inhibitor of PKR. The authors should demonstrate that the inhibitor was necessary and has worked, for example by documenting eIF2alpha or PKR phosphorylation after in vivo mRNA transfection in the absence of C16. In addition, the intratracheal mRNA delivery experiments could be performed in PKR knock out mice where the method should work without the inhibitor.*

Response to reviewer: We tested the effect of delivery of aPali mRNA with and without C16 in the lungs of mice. We infected the mice one day later and analyzed the lungs by plaque assay, finding that viral titers were increased in both cases (Supplemental Figure 7). However, the difference between the infected and mock mice, when C16 was included, was exacerbated. Because mice have limitations as a model for RSV infections, we used C16 due to the larger differences in RSV replication in the positive and negative controls. This point was made clearer in the text (Results, Optimizing mRNA delivery..., "Because RSV does not replicate...").

3. *The expression level of a number of cytokines was determined in mouse lungs sampled from animals that had been treated with in vitro transcribed mRNA, subsequently infected with RSV and sacrificed on day 4 after infection. It is important to evaluate cytokine expression in lung samples at 8 or 24 hours after mRNA transfection, prior to RSV challenge. This could support the statement that the mRNA is poorly inflammatory in vivo.*

Response to reviewer: We transfected mRNA, diluted in water, into the lungs of mice, sacrificed the mice and harvested the lungs 24 h later, and evaluated the expression of several pro-inflammatory cytokines by ELISA in lung homogenates. We found that there was no relevant increase in concentration in any of the measured cytokines (Supplemental Figure 11).

4. *There is only a minor difference in RSV neutralization activity between Pali and aPali mRNA. It would be very interesting to make a comparison of the RSV inhibitory activity of VHH with aVHH mRNA in vivo where the advantage of anchoring over the rapidly diffusing non-anchored VHH would perhaps be more pronounced.*

Response to reviewer: We transfected mRNA encoding for either aPali, sPali, RSV aVHH, or RSV sVHH into the lungs of mice and infected them with RSV 7 days later (an increase from the 1 day experiments originally included). At 4 dpi, we sacrificed the mice and harvested the lungs before evaluating the replication of RSV by plaque assay (Figure 7a) and qRT-PCR (Figure 7b). Interestingly, we found that both aPali and sPali significantly inhibited RSV replication, likely due to the longer half-life of the whole antibody (~28 days in serum). However, we found that while the anchored VHH did significantly inhibit RSV, the secreted VHH did not, indicating that the anchor was preventing diffusion of the VHH away from the target epithelial surface of RSV.

5. *An important potential advantage of the authors' approach is that mRNA delivery of an anchored antibody could result in more sustained protection against RSV challenge. It would therefore be interesting to test in vitro and in vivo how long after transfection of aPali/aVHH compared with sPali/sVHH protection lasts. Thus start infections at 2 days or longer after mRNA administration.*

Response to reviewer: As mentioned above, we compared the RSV inhibiting ability of both aPali and sPali as well as aVHH and sVHH, finding no difference between the secreted and anchored forms of the whole antibody but a significant difference between the secreted and anchored forms of the VHH, as measured by plaque assay and qRT-PCR (Figure 7a and b).

6. *The number of animals that were used is low and does not allow to conclude that there are significant differences between the groups based on statistical power. E.g. in Figure 6g, there appears to be no significant difference between RSV and aVHH + RSV. It is highly recommended to document the protection based on repeat experiments.*

Response to reviewer: A significant difference was recorded between the groups in Figure 6G by Kruskal-Wallis (which has more power than a one-way ANOVA due to no assumption of variance or normality in the groups). We also completed another trial during testing of the secreted and anchored VHH and Pali at 7 days, again showing a significant decrease in RSV replication by both plaque assay and qRT-PCR (again using Kruskal-Wallis).

7. *The PFU graph in figure 4b indicates a 4-8 fold difference in the number of plaques/g lung between the saline and jet PEI vs the VR and naked setup. The tissue slides in figure 4d suggest that the naked mRNA setting had no detectable RSV foci. Please explain this difference.*

Response to reviewer: This discrepancy is explained by our protocol for plaque assay. We homogenize the lung tissue (without perfusion or lavage), centrifuge briefly to collect the tissue at the bottom of the tube, and use the supernatant from this preparation to inoculate the plaque assays. Because of this, the plaque assay is a measure of *free RSV virions*. The tissue staining, on the other hand, is detecting *cell-associated RSV* that has

replicated substantially for visualization with the panRSV antibody staining. This discrepancy is also why we opted to utilize both plaque assay and qRT-PCR as measurements of RSV replication for the lung work. These points were made clearer in the text (Results, Optimizing mRNA delivery ..., “The reduced staining ...”).

8. *The right hand panel of figure 1c and the top right hand panel of figure 1d essentially show the same: Vero cells transfected with aPali mRNA and 24h later permeabilized and stained with an anti-human antibody. Yet the outcome looks very differently. Please explain this apparent discrepancy.*

Response to reviewer: This discrepancy is explained by a difference in contrast enhancement between the two experiments. In figure 1c, all images were taken at 24h, after most aPali had reached the cell surface. Because of this, the brightest intensity was lower than in Figure 1d, and as such the white point was set at a lower value. In figure 1d, the ER was stained very brightly at the early time points, such that single pixels had much higher intensities than after the aPali had reached the surface of the cell. Thus, the white point was set to a higher value. This point was made clearer in the text (Results, aPali is anchored ..., “Note that because the aPali ...”).

9. *In figure 3e, lung cells from 1 out of 3 “transfected” mice did not score V5-positive. Does this reflect variability in the in vivo efficiency of the mRNA delivery? Please explain.*

Response to reviewer: The low staining in the one mouse was likely due to variability in the intratracheal aerosol delivery. This point was clarified in the text (Results, Lung distribution..., “One of the three mice...”).

10. *Results: ... a significant fraction of it was likely free in the cytosol. Why was this mRNA not detected (cfr. Figure 3b)?*

Response to reviewer: The only way to detect free mRNA in the cytosol is by taking the total cellular mRNA signal, and subtracting mRNA signal that is trapped in endosomes (see reference 30, Kirschman et. al.). Because detected mRNA was not trapped in the parts of the endosomal system we specifically stained for, by extension we concluded that some of the mRNA was free in the cytosol. This point was clarified in the text (Results, Lung distribution..., “Since 75.6%...”).

11. *It is unclear how the RT-qPCR for RSV F RNA was normalized. Please explain in the methods section.*

Response to reviewer: In order to perform absolute quantification of RSV F mRNA quantity using RSV F gene standards, we normalized the amount of RNA (250 ng) used for cDNA synthesis and used the same amount of cDNA for qRT-PCR. This was made clear in the methods section (Methods, qRT-PCR, “In order to perform absolute...”).

12. *The authors should document that the mRNA derived sPali is as potent (anti-RSV activity) as the licensed recombinant humanized antibody counterpart i.e. Palivizumab.*

Response to reviewer: This was an important point that we had originally overlooked. We have discussed this result in the text (Discussion, Paragraph “While previous studies...”, “Critically, delivery of...”).

13. *There is a risk that aPali produced in limited amounts on a mRNA transfected cell surface can enhance RSV virion attachment and subsequent entry. The authors should test an in vitro dose range of mRNA to assess this possibility.*

Response to reviewer: We performed a titration of aPali mRNA from 100ng to 2000ng per well in a 24-well plate. We then analyzed the transfected cells by plaque assay (Supplemental Figure 3). At low concentrations of mRNA, we merely measured a dose response, not enhanced infection, likely due to either a reduced number of transfected cells or significantly reduced coverage on transfected cells. These points were discussed in the text (Results, aPali is anchored ..., “To verify that transfection ...”).

14. *The authors should test in a human cell line, e.g. A549 (human lung epithelial-derived cell line), that aPali can also be prevent RSV infection.*

Response to reviewer: We tested aPali and sPali transfection and RSV inhibition by microscopy in A549 cells (Supplemental Figure 1). Additionally, we tested the RSV aVHH and Flu aVHH (as an mRNA control) transfection and RSV inhibition by microscopy in A549 cells (Supplemental Figure 12).

15. *X-axis labeling in 6d should be aVHH instead of VHH to concord with the description in the legend.*

Response to reviewer: All figures were updated to include the new nomenclature of RSV aVHH (to differentiate from the Flu aVHH control mRNA).

16. *Last paragraph of the results: ... we delivered ... of aVHH or aPali mRNA by ... In figure 6e-f only aPali is shown.*

Response to reviewer: The appropriate sentence was updated with the correct text.

17. *Last paragraph of the introduction. ... expressed anchored neutralizing antibodies,... (3) were minimally immunogenic. “Immunogenic” is most often used to denote that something induces B or T cell responses. Such responses were not studied in the manuscript. Presumably the authors mean minimally inflammatory. However, it is not clear if this is the case cfr. remarks 2 and 3 above.*

Response to reviewer: The appropriate sentence was updated with the correct text.

18. *Methods, plaque assay: “mRNA knockdown” is commonly used for si- or shRNA mediated reduction of cellular mRNA.*

Response to reviewer: The appropriate sentence was updated with the correct text.

19. *Results, paragraph “Optimizing mRNA...”: “To compare the therapeutic...” A therapeutic experiment, meaning treatment of already RSV-infected animals, is not described. Please adapt the statement.*

Response to reviewer: The appropriate sentence was updated with the correct text.

Reviewer 2:

1. *Authors expressed a humanized antibody in healthy mice, which can activate foreign body immune response and subsequently be a reason of resistance to the virus. Thus, a control with non-specific humanized antibody mRNA (or at least a scrambled mRNA) is needed.*

Response to reviewer: We designed an mRNA encoding a GPI-anchored VHH antibody that neutralizes influenza virus (from Reference 46), termed Flu aVHH. We tested Flu aVHH in vitro, using microscopy (Figure 6A and B, and Supplemental Figure 12) and plaque assay (Supplemental Figure 13), and in vivo, using qRT-PCR (Figure 7a) and plaque assay (Figure 7b). The Flu aVHH did not have significant effects on RSV in vitro and in vivo.

2. *Fig 4. 1) Standard deviations for certain groups are big; 2) We would expect to see standard deviation of similar size for different groups in the same experiment unless there is a certain reason for observation of an increased variance in certain groups. I suggest, that the experiments should be repeated with a higher animal number per group.*

Response to reviewer: A significant difference was recorded between the groups in several experiments by Kruskal-Wallis (which has more power than a one-way ANOVA due to no assumption of variance or normality in the groups). We also completed another trial during testing of the secreted and anchored VHH and Pali at 7 days, again showing a significant decrease in RSV replication by both plaque assay and qRT-PCR (again using Kruskal-Wallis).

3. *In the paper authors study immune response 5 days after mRNA injection. It has been shown in several studies (Mol. Ther. 20 (2012) 948-953; Biomaterials 109 (2016) 78-87) that immune response is usually gone 72h after mRNA injection, thus I suggest to check immune response markers after 6-24h.*

Response to reviewer: We transfected mRNA, diluted in water, into the lungs of mice, sacrificed the mice and harvested the lungs 24 h later, and evaluated the expression of several pro-inflammatory cytokines by ELISA in the lung homogenates. We found that there was no relevant increase in concentration in any of the measured cytokines (Supplemental Figure 11).

4. *It would be helpful if authors describe explicitly what they mean by Mock, Saline 1 and Saline 2 (Fig.4); add names of the cell lines to the descriptions for fig 1; show a picture of Vero cells infected with the virus uptaken by the cell instead of the bad resolution picture.*

Response to reviewer: We updated the text in Figure 4 for clarity and in the associated text in the results section. Saline 1 and Saline 2 represent two different untransfected, but RSV infected, mice. This point was made clearer in the text (Results, Optimizing mRNA delivery..., “All three delivery vehicles...”). Cell line names were added to all relevant captions. The infected cells in Figure 1e have all of the structures typical of a late-stage RSV infection, including: (1) inclusion bodies (the large round structures in RSV N staining (green), and (2) RSV filaments, the long thin structures stained by both RSV N (green) and the panRSV antibody (blue).

Reviewers' comments:

Reviewer #1 (Remarks to the Author):

The authors report on an mRNA based expression approach for antibodies or single domain antibodies that can neutralize human RSV. In addition to the transient expression of normal antibody constructs, also GPI-anchored variants are produced, which seem to perform better at preventing RSV infection. A PKR inhibitor was added to the mRNA in almost all in vivo experiments. It is impressive that 28 days after intracheal administration of naked mRNA coding for a GPI anchored VHH, that VHH is still detectable in the mouse lungs.

Major remarks

1. Transfection with a conventional irrelevant control antibody for palivizumab is missing and should be included in the experiments.
2. C16, which reportedly inhibits PKR, was co-delivered with the IVT mRNA in all experiments, except for the experiments of which the results are shown in Suppl. Fig 8 and 12. It is interesting that treatment of mice with C16 24h prior to RSV infection significantly increases the amount of RSV RNA on day 1 after infection. However, 2 important questions remain unanswered: 1) did the inhibitor work? 2) To what extent does C16 affect the innate responses against the mRNA in vivo? Such responses are known to have an antiviral effect. The authors should document that the inhibitor is specific for PKR, for example by showing that PKR is activated in vivo by the mRNA treatment (the prototypical substrate of PKR is eIF2alpha and there are good assays available to quantify eIF2alpha phosphorylation) and that C16 inhibits this. Alternatively, the effect of C16 on the antiviral effect should be tested in PKR ko mice, which are readily available, in which case the inhibitor should not have an effect.

Supplementary Fig. 8c partially addresses the C16 effect on the prophylactic aPali mRNA anti-RSV effect but unfortunately lacks a negative control, i.e. an mRNA coding for an irrelevant GPI-anchored IgG. In addition, the relevant statistical comparisons that are necessary to conclude that the mRNA-aPali reduces RSV replication are: saline C16 RSV vs aPali C16 RSV; saline wo C16 RSV vs aPali wo C16 RSV and, most importantly a comparison with a-irrelevant IgG with or without C16 and RSV.

3. Figure 4a lacks an irrelevant mRNA IgG control. The statistics in 4e allow to conclude that Pali IM (the standard of prophylactic care in at risk infants) is not significantly different from any group. Yet the authors write that "remarkably ...mRNA was more effective ... than palivizumab given im."

4. Fig. 5a lacks an mRNA encoding an irrelevant IgG control.

5. It is very surprising that RSV infection does not induce a change in CCL5, IL12p70, CCL-3 and IL6 compared to non-infected mice (Suppl. Fig. 11). How can this be explained? Suppl. Fig 12 shows that aVHH mRNA application is associated with a significant increase in CCL5 and CCL3, yet the authors write that there is no "biologically" significant increase.

6. Figure 6d, e and f lack the Flu aVHH + RSV control.

7. Figure 7a,b. It is interesting to see that the number of RSV plaques and RSV RNA copy numbers are statistically significantly higher compared to the mock for the non-treated RSV group, the RSV sVHH RSV group and the FluVHH RSV group. However, the question is whether there is a significant difference between these values in the mice that were treated

with sPali, aPali, RSV sVHH or RSV aVHH and the Flu VHH RSV group. In other words is there a statistically significant difference in RSV replication between the negative control RNA (Flu VHH) and RSV a/aPali and RSV VHH treated mice?

8. Discussion: "... did not alter base line cytokine levels". The chemokines CCL5 and CCL3 are, however, significantly higher in the aVHH group compared with the control group (suppl. Fig. 12).

Other remarks:

1. Please use page and line numbering.
2. The last 2 sentences of the abstract are speculative and should be avoided in the abstract.
3. Figure 1g: it is unclear on how many samples the boxes are based.
4. Legend figure 2b: what does "individual viral particles were isolated in software mean"? How can a physical particle be isolated in software?

Reviewer #2 (Remarks to the Author):

The study explore an original idea of anchoring IgG in lung epithelial cells instead of secreting it into intracellular space. The paper provides a good understanding of IgG mRNA expression, transport and assembly inside the cell.

After the last review the authors improved the paper and addressed some of reviewer's comments. However, there are still a few important issues to be addressed:

From the presented data on fig 4 and 7 it is not clear whether the flu VHH mRNA effects on RSV PFU/g and copy number are statistically different from 1) RSV infection treatment and 2) Pali and RSV VHH treatment in order to claim that non-specific mRNA doesn't have anti-viral effect. Additional statistical analysis has to be done.

Reviewer comments are in bolded font. Responses are below each point in regular weight font.

REVIEWER 1

Major remarks

1. Transfection with a conventional irrelevant control antibody for palivizumab is missing and should be included in the experiments.

In order to include an irrelevant control for each experiment, we transfected mice with mRNA encoding for either Flu-aVHH or anchored Pali heavy chain *only*. The IgG heavy chain only mRNA was included as a control because it includes sequences from our aPali construct, but without the light chain, and is not inhibitory. We feel this is a better control than a random antibody which may have undiscovered RSV inhibitory effects. Infected controls were delivered saline only. One day later, all animals were infected with RSV L19. At four days post infection, all animals were sacrificed and their lungs were harvested and processed for qRT-PCR for RSV F copy number.

In order to include these results in relevant graphs, PCR data was normalized to the RSV F copy number of the RSV only (saline) controls. This means that all adjusted qRT-PCR graphs will have an updated y-axis label of “Normalized RSV F Copy Number” with the appropriate range.

2. C16, which reportedly inhibits PKR, was co-delivered with the IVT mRNA in all experiments, except for the experiments of which the results are shown in Suppl. Fig 8 and 12. It is interesting that treatment of mice with C16 24h prior to RSV infection significantly increases the amount of RSV RNA on day 1 after infection. However, 2 important questions remain unanswered: 1) did the inhibitor work? 2) To what extent does C16 affect the innate responses against the mRNA in vivo? Such responses are known to have an antiviral effect. The authors should document that the inhibitor is specific for PKR, for example by showing that PKR is activated in vivo by the mRNA treatment (the prototypical substrate of PKR is eIF2alpha and there are good assays available to quantify eIF2alpha phosphorylation) and that C16 inhibits this. Alternatively, the effect of C16 on the antiviral effect should be tested in PKR ko mice, which are readily available, in which case the inhibitor should not have an effect.

Supplementary Fig. 8c partially addresses the C16 effect on the prophylactic aPali mRNA anti-RSV effect but unfortunately lacks a negative control, i.e. an mRNA coding for an irrelevant GPI-anchored IgG. In addition, the relevant statistical comparisons that are necessary to conclude that the mRNA-aPali reduces RSV replication are: saline C16 RSV vs aPali C16 RSV; saline wo C16 RSV vs aPali wo C16 RSV and, most importantly a comparison with a-irrelevant IgG with or without C16 and RSV.

From our previous experiments, we had lung tissue samples from saline control animals and animals transfected with aPali mRNA, either with or without C16. These four groups were all infected with RSV. One set of animals neither transfected nor infected, were used as a mock control group.

In order to determine the effect of C16 (PKR inhibitor) on innate responses, we assayed these lung samples for eIF2 α phosphorylation (the consequence of PKR activation), by western blot and ELISA. For the western blot, we exposed a cell line (human foreskin fibroblasts) with

sodium arsenite to induce eIF2 α phosphorylation as a positive control. We stained the blot for both total and phosphorylated eIF2 α . A second blot was used for GAPDH staining as a sample processing control. From these experiments, we can conclude that C16 did not have a significant (or even detectable) effect on PKR activation. There was a concern initially that mRNA and the infection may cause excessive inflammation in the lung, hence our inclusion of C16 in subsequent experiments. It should be clear though, that such inflammation was never observed.

Given the data mentioned above, we feel strongly that it is unwarranted to include an additional irrelevant antibody control while only testing the effects of C16 on the immune response and RSV inhibition efficacy of mRNA expressed palivizumab. The irrelevant antibody controls are included later in the manuscript in the subsequent RSV challenge experiments where C16 was used.

In addition, given that C16 does not exhibit any inhibitory effects on the virus, our IACUC will not allow additional animals to be added exclusively for control purposes with C16, thus we could not include an additional irrelevant IgG control in this instance. The irrelevant antibody controls are included later in the manuscript in the subsequent RSV challenge experiments where C16 was used.

Text was added reflecting these additions (Page 4, Line 195).

We specifically excluded C16 from cytokine response experiment (supplemental figure 12) to rule out any extreme immune response to aPali, sPali and RSV-aVHH at day 1 post-delivery. Given the results above regarding eIF2 α phosphorylation, we did not deem it necessary to include C16 in this experiment.

3. Figure 4a lacks an irrelevant mRNA IgG control. The statistics in 4e allow to conclude that Pali IM (the standard of prophylactic care in at risk infants) is not significantly different from any group. Yet the authors write that “remarkably ...mRNA was more effective ... than palivizumab given im.”

In Figure 4a, we were testing the effect of delivery vehicles on the efficacy of aPali mRNA to prevent RSV infection. Therefore, we did not include an irrelevant mRNA control. The control for the condition we utilized going forward to assess overall efficacy can be seen in Fig 5. This control was at the highest dose of mRNA used and therefore there was no need to repeat the control at lower doses. Our IACUC did not feel that it was justified to sacrifice more animals for controls at lower doses.

The results regarding Figure 4e was rephrased appropriately (Page 5, Line 230).

4. Fig. 5a lacks an mRNA encoding an irrelevant IgG control.

We added aPali HC (heavy chain) only and Flu aVHH mRNA controls to the PCR data in Figure 5b. As discussed above, this data was normalized in order to make comparisons to our previous experiments. We also compared all groups to both the RSV only and to two irrelevant mRNA controls (aPali HC only and Flu aVHH mRNAs) in the statistical analyses and updated the graph and associated text in the results (Page 5, Line 245) accordingly.

5. It is very surprising that RSV infection does not induce a change in CCL5, IL12p70, CCL-3 and IL6 compared to non-infected mice (Suppl. Fig. 11). How can this be

explained? Suppl. Fig 12 shows that aVHH mRNA application is associated with a significant increase in CCL5 and CCL3, yet the authors write that there is no “biologically” significant increase.

To address the concern that RSV infection does not induce a cytokine response, we cited Moore et. al. (2009) in the main text, Reference 31, which found that maximal cytokine expression by PCR was measured at 8 days post infection, with no changes at 4 days post-infection. We sacrificed animals at 4 days post infection, when plaque assay titers and viral RNA amounts were the highest. Appropriate text was added in the Results section (Page 5, Line 255).

We rephrased the results, stating that CCL5 and CCL3 were increased with aVHH mRNA, but not with aPali or sPali mRNA (Page 6, Line 264).

6. Figure 6d, e and f lack the Flu aVHH + RSV control.

As discussed above, normalized irrelevant mRNA control data was added to the *in vivo* PCR graph of Figure 6 and all data was normalized in order to make comparisons across experiments. We also compared all groups to both the RSV only and to two irrelevant mRNA controls (aPali HC only and Flu aVHH mRNAs) in the statistical analyses and updated the graph accordingly. The appropriate text in the results was added (Page 6, lines 301-304).

7. Figure 7a,b. It is interesting to see that the number of RSV plaques and RSV RNA copy numbers are statistically significantly higher compared to the mock for the non-treated RSV group, the RSV sVHH RSV group and the FluVHH RSV group. However, the question is whether there is a significant difference between these values in the mice that were treated with sPali, aPali, RSV sVHH or RSV aVHH and the Flu VHH RSV group. In other words is there a statistically significant difference in RSV replication between the negative control RNA (Flu VHH) and RSV a/aPali and RSV VHH treated mice?

In order to address this concern, we performed our statistical analyses between all groups. When we analyzed the viral titers, we found that sPali, aPali, RSV sVHH, and RSV aVHH all significantly reduced titer compared to the RSV only group. When compared to the Flu aVHH control, all treatments except for RSV sVHH significantly inhibited production of infectious RSV progeny.

When we analyzed the quantity of RSV F copy number, we found that sPali, aPali, and RSV aVHH, but not RSV sVHH, significantly inhibited RSV RNA production when compared to the RSV only group. We also observed significant differences between the negative control mRNA (Flu aVHH) and sPali and RSV aVHH mRNA by qRT-PCR. We calculated a significance level of $p=0.06$ between aPali and Flu aVHH mRNA treated animals, suggesting that aPali also has an effect on RSV F copy number at 7 days post transfection. We have updated the graphs for Figure 7a and b and the associated text in the results (Page 7, Lines 313-321).

It is important to note here that although aPali has a significance level of only $p=0.06$ compared to Flu-aVHH in qPCR, it is significantly different in the plaque assay which is a biologically relevant assay in terms of viral pathogenesis and disease.

8. Discussion: “... did not alter base line cytokine levels”. The chemokines CCL5 and CCL3 are, however, significantly higher in the aVHH group compared with the control group (suppl. Fig. 12).

This line was corrected in the discussion (Page 7, Line 339).

Other remarks:

1. Please use page and line numbering.

We added page and line numbering.

2. The last 2 sentences of the abstract are speculative and should be avoided in the abstract.

We rephrased the ending of the abstract.

3. Figure 1g: it is unclear on how many samples the boxes are based.

We plotted individual plaque assay data from this experiment.

4. Legend figure 2b: what does “individual viral particles were isolated in software mean”? How can a physical particle be isolated in software?

We described this process in more detail. Specifically, single viral particles were identified using the N staining and a 1 μm square region of interest (ROI) in the Vutara SRX software. See line 693, page 14 for the description in the Methods section.

REVIEWER 2

From the presented data on fig 4 and 7 it is not clear whether the flu VHH mRNA effects on RSV PFU/g and copy number are statistically different from 1) RSV infection treatment and 2) Pali and RSV VHH treatment in order to claim that non-specific mRNA doesn't have anti-viral effect. Additional statistical analysis has to be done.

To address these concerns, we performed statistics between all groups rather than just against the mock infected animals. Asterisks were only added if the difference between the groups identified by the bars was significant ($p < 0.05$).

Reviewers' comments:

Reviewer #1 (Remarks to the Author):

Major comments:

Figure 4e lacks a group with delivery into the lungs of an mRNA coding for an irrelevant control IgG (or the heavy chain of palivizumab, which seems to be preferred by the authors as an irrelevant control). Furthermore, in the text it is stated that " 100 µg of aPali mRNA significantly reduced RSV F copy number by 90% compared to the RSV control animals, while 20 µg of aPali mRNA yielded no significant decrease in RSV F copy number (Fig. 4e)". In the figure legend the 0.05 probability calculation is indicated for the saline and 20 ug mRNA, not for the 100 ug group. Which groups were compared exactly?

Figure 6f and 6g are both based on the experiment of which the result is also shown in Fig6 e. Yet in 6f there are 3 groups and in 6g 5 groups are compared.

Figure 5a lacks a negative control mRNA. In figure 5b, which presents data from the same experiment, additional groups pop up.

Suppl. Figure 4 lack a control transfection with an irrelevant IgG encoding mRNA.

In Suppl. Figure 14 a secreted RSV-specific VHH construct is compared with an anchored Flu-specific VHH. Comparison should be done between 2 secreted VHHs that have different specificities or, between secreted and anchored for the same specificity.

Reviewer comments are in bolded font. Responses are below each point in regular weight font.

REVIEWER 1

Major remarks

1. Figure 4e lacks a group with delivery into the lungs of an mRNA coding for an irrelevant control IgG (or the heavy chain of palivizumab, which seems to be preferred by the authors as an irrelevant control). Furthermore, in the text it is stated that “100 µg of aPali mRNA significantly reduced RSV F copy number by 90% compared to the RSV control animals, while 20 µg of aPali mRNA yielded no significant decrease in RSV F copy number (Fig. 4e).” In the figure legend the 0.05 probability calculation is indicated for the saline and 20 µg mRNA, not for the 100 µg group. Which groups were compared exactly?

In order to include an irrelevant control for this experiment, we transfected mice with mRNA encoding for Flu-aVHH (Results, lines 230-232). One day later, all animals were infected with RSV L19. At four days post infection, all animals were sacrificed and their lungs were harvested and processed for RT-PCR for RSV F copy number.

In order to include these results in relevant graphs, PCR data was normalized to the RSV F copy number of the RSV only (saline) controls. This means that the adjusted qRT-PCR graph will have an updated y-axis label of “Normalized RSV F Copy Number” with the appropriate range. The statistics and significantly different groups were updated in the caption. All groups were compared in the updated graph.

2. Figure 6f and 6g are both based on the experiment of which the result is also shown in Fig 6 e. Yet in 6f there are 3 groups and in 6g 5 groups are compared.

Plaque assay titers from the Flu aVHH and Pali HC only treated groups were included in Figure 6f. In order to include these results in the relevant graphs, plaque assay titers were normalized to the RSV only group, in the same way as the PCR data above (Results, lines 298 and 305). Thus, the updated graph will have a y-axis label of “Normalized PFU/g Lung Tissue”. The caption was updated as appropriate.

3. Figure 5a lacks a negative control mRNA. In figure 5b, which presents data from the same experiment, additional groups pop up

Plaque assay titers from the Flu aVHH and Pali HC only treated groups were included in Figure 5a (Results, lines 242-245). Normalization was performed as described above. The caption was updated as appropriate.

4. Suppl. Figure 4 lack a control transfection with an irrelevant IgG encoding mRNA.

Cells treated with Flu aVHH mRNA were infected with RSV and plaque assays were performed. To account for differences in RSV stock viral titers, we normalized the titers to the control (0 ng) mRNA condition (Results, lines 126-127). Supplementary Figure 4 and the caption were updated accordingly.

5 In Suppl. Figure 14 a secreted RSV-specific VHH construct is compared with an anchored Flu-specific VHH. Comparison should be done between 2 secreted VHHs that have different specificities or, between secreted and anchored for the same specificity.

Cells treated with Flu sVHH mRNA were infected with RSV and plaque assays were performed and plotted in Supplementary Figure 14a (Results, lines 291-292). The caption was updated as appropriate.

REVIEWERS' COMMENTS:

Reviewer #1 (Remarks to the Author):

The authors have done efforts to add the relevant controls in the experiments.

One minor comment: the statistics indicated in figure 6g (one asterisk + one bracket) does not appear to be in line with the statements in lines 306-308.

Reviewer comments are in bolded font. Responses are below each point in regular weight font.

REVIEWER 1

1. One minor comment: the statistics indicated in figure 6g (one asterisk + one bracket) does not appear to be in line with the statements in lines 306-308.

We updated Figure 6g with individual asterisks to more clearly indicate significant differences.